# From Mechanisms to Medicine: Neurovascular Coupling in the Diagnosis and Treatment of Cerebrovascular Disorders: A Narrative Review

**DOI:** 10.3390/cells14010016

**Published:** 2024-12-27

**Authors:** Lu Yang, Wenbo Zhao, Yuan Kan, Changhong Ren, Xunming Ji

**Affiliations:** 1Department of Neurology, Xuanwu Hospital, Capital Medical University, Beijing 100053, China; yanglu_medical1991@163.com (L.Y.); zhaowb@xwh.ccmu.edu.cn (W.Z.); kanyuan1701087@163.com (Y.K.); 2Beijing Institute of Brain Disorders, Capital Medical University, Beijing 100054, China; 3Beijing Key Laboratory of Hypoxic Conditioning Translational Medicine, Xuanwu Hospital, Capital Medical University, Beijing 100053, China; 4Department of Neurosurgery, Xuanwu Hospital, Capital Medical University, Beijing 100053, China

**Keywords:** neurovascular coupling, functional hyperemia, cerebrovascular diseases, neurovascular unit, cerebral blood flow

## Abstract

Neurovascular coupling (NVC) refers to the process of local changes in cerebral blood flow (CBF) after neuronal activity, which ensures the timely and adequate supply of oxygen, glucose, and substrates to the active regions of the brain. Recent clinical imaging and experimental technology advancements have deepened our understanding of the cellular mechanisms underlying NVC. Pathological conditions such as stroke, subarachnoid hemorrhage, cerebral small vascular disease, and vascular cognitive impairment can disrupt NVC even before clinical symptoms appear. However, the complexity of the underlying mechanism remains unclear. This review discusses basic and clinical experimental evidence on how neural activity sensitively communicates with the vasculature to cause spatial changes in blood flow in cerebrovascular diseases. A deeper understanding of how neurovascular unit-related cells participate in NVC regulation is necessary to better understand blood flow and nerve activity recovery in cerebrovascular diseases.

## 1. Introduction

In the intricate network of the human brain, the precise coordination between neural activity and vascular response, called neurovascular coupling (NVC), is essential for maintaining optimal brain function and health [1]. This dynamic interplay between neural activity and vascular response unfolds in every corner of the brain. When neurons in a specific area are active, they can trigger vasodilation directly by stimulating vascular smooth muscle cells (VSMCs) and pericytes or indirectly through astrocytes, thus rapidly increasing local cerebral blood flow (CBF) [1]. This response is not only swift but also precise and is a crucial component of brain health regulation [2]. However, this regulatory mechanism is often impaired in various cerebrovascular diseases such as acute stroke, subarachnoid hemorrhage (SAH), cerebral small vascular disease (CSVD), and vascular dementia, leading to brain dysfunction and neuronal damage [3,4,5]. Research on NVC not only helps us understand the mechanisms of automatic regulation of CBF but is also crucial for revealing the pathophysiological mechanisms of cerebrovascular diseases.

Significant advances in brain imaging technology and cellular molecular biology in recent years have enhanced our understanding of NVC. Even if there is a small change in the interaction between cells around the cerebral microcirculation blood vessels, NVC will be affected, causing brain function damage. Functional reconstruction of NVC not only promotes cell survival and cell debris removal but also promotes brain microcirculation reconstruction and antagonizes ischemic and hemorrhage-related brain injury [6]. Therefore, the mechanism study of NVC can reveal the abnormal brain functional activity and its correlation with nerve function injury earlier and more sensitively. This approach allows for a holistic treatment of neurological diseases and offers innovative strategies to ameliorate compromised neurological symptoms. In general, the evaluation of NVC function before the disease is helpful for early diagnosis and timely intervention, while the evaluation of NVC after the disease helps judge the severity and prognosis of the disease.

This review will delve into the cellular mechanisms of NVC, exploring how different cellular components contribute to its function and examining the changes and clinical implications of these mechanisms in common cerebrovascular diseases. By synthesizing the latest research findings and methodologies, this review aims to provide a comprehensive overview of the mechanisms of NVC and its dual role in maintaining brain health and responding to pathological states, providing its potential targets in developing future strategies for the treatment and prevention of cerebrovascular diseases.

## 2. Overview of Neurovascular Coupling

NVC is a crucial physiological process that ensures CBF is finely tuned to neuronal activity, which is essential for maintaining optimal brain function. The anatomical foundation of NVC begins with the circle of Willis, which distributes blood through cerebral arteries that traverse around the pia meningeal, forming the meningeal artery. These arteries then penetrate deeper into the brain parenchyma, transitioning into deep penetrating arterioles while establishing perivascular spaces between the initial cortical penetration and the pia. As arterioles delve further, their walls integrate with the pia, the perivascular spaces close off, and the vessels branch into terminal capillaries. This complex vascular architecture ensures that every neuron is close to blood vessels, facilitating an efficient exchange of nutrients and waste products, which is foundational for brain health [2,7,8].

Functionally, the neurovascular unit (NVU) operates as the core component of NVC. Comprising neurons, astrocytes, VSMCs, and pericytes, the NVU orchestrates the regulation of CBF (Figure 1). Neurons, either directly or through astrocytes, signal VSMCs and pericytes to dilate nearby blood vessels. This response increases blood flow precisely where it is most needed to support the metabolic demands of active brain regions [4]. Such rapid and precise vascular responses to neural activity have been documented to occur within seconds and can affect areas several millimeters from the site of neuronal activation [2,9].

Moreover, NVC involves a sophisticated multicellular response that adjusts local microvascular flows and influences upstream larger arteries (Figure 2). This advanced coordination is critical for the effectiveness of modern brain imaging techniques, which rely on vascular signals as proxies for neuronal activity [1,10]. The brain’s limited energy reserves highlight the essential nature of this mechanism; a disruption in NVC can quickly halt brain function and lead to irreversible damage within minutes, underscoring the urgency in maintaining NVC integrity [7].

Given its pivotal role in the automatic regulation of CBF and its involvement in the pathophysiology of various cerebrovascular diseases such as acute stroke, SAH, CSVD, and vascular dementia, NVC has garnered significant research attention. Enhancing the understanding of NVC not only illuminates its fundamental role in brain health but also opens avenues for developing targeted therapeutic strategies for these debilitating conditions. This underscores NVC’s profound significance in maintaining cerebral health and addressing cerebrovascular diseases.

## 3. Cellular Contributions to Neurovascular Coupling

We next elaborate on the mechanism of action of each cell type within the NVU in NVC, which displays selected representative studies of cellular mechanisms of NVC, mainly showing the cells involved in studying NVC, animal models, stimulation modalities, and imaging techniques.

### 3.1. Neuron and NVC

The intricate interplay between vascular and neuronal functions is one of the most defining characteristics of cerebral circulation, serving as the foundation for functional imaging signals [10]. Specifically, cortical neural activity is intricately linked to both the local arterial diameter and CBF [11]. Nonetheless, our understanding of the neurons responsible for regulating the dynamics of cerebral arteries remains limited. The vascular responses triggered by excitatory neurons result in substantial increases in neural activity and oxygen consumption but not interneurons [12]. This, in conjunction with the well-documented proximity of microvasculature to interneurons and their extensions, implies the existence of distinct mechanisms through which neural activity modulates cholinergic signaling in the basal forebrain [13]. Employing the NVC response to various sensory stimuli—including whisker, forepaw, hind paw, and visual stimuli—researchers have conducted sophisticated experiments that show how the dispersion of hemodynamic activity precisely mirrors the neural response, which is propelled by synaptic activity arising from intracortical processing [14,15,16].

Furthermore, recent research has shed light on the specific types of neurons and their mediators involved in NVC. Lecrux et al. conducted a study that identified neurons activated by whisker stimulation in the rat somatosensory cortex, employing double immunohistochemistry techniques for c-Fos alongside markers for glutamatergic and gamma-aminobutyric acid (GABAergic) neurons. Their findings revealed that pyramidal neurons serve as “neurogenic hubs” within the context of NVC [17]. Another study has shown that the hyperemic response to basal forebrain stimulation is predominantly mediated by glutamate released from activated pyramidal cells. This process results in the release of vasoactive PGE2 and EETs, which are likely produced by activated astrocytes. Furthermore, it has been observed that whisker-evoked changes in CBF and local field potential (LFP) responses are influenced by the tone of acetylcholine (ACh) [18]. LFP refers to weak electrical signals generated by the synchronized activity of many neurons in the cerebral cortex or neural tissue. These electrical signals are generated by the superposition of postsynaptic potentials (mainly excitatory and inhibitory postsynaptic potentials) of neurons, and they reflect the activity state of the neuronal population.

Inhibitory GABA interneurons play a critical role in NVC, primarily by modulating the output of excitatory pyramidal cells, which function as the central neuronal mediators of the hemodynamic response, either directly or through signaling by astrocytes [19]. The neuronal activity was consistently affected by the inhibition of glutamate and GABA-A receptors, which resulted in a reduction of postsynaptic activity contributions. While hemodynamic changes continued to occur, they were somewhat diminished [20]. Furthermore, the blood flow and volume changes induced by photostimulation were reduced by 75 to 80% in the presence of a NOS inhibitor, indicating that inhibitory neurons regulate blood flow through the action of NO [21].

Echagarruga et al. investigated the cellular mechanisms that regulate the baseline and stimulus-induced dilation of cortical arteries in conscious, head-immobilized mice [22]. The results indicated that it is the activity of neurons expressing nNOS, rather than solely the activity of pyramidal neurons, that influences the alterations in both resting and reactive arterial dimensions [23]. A systematic review that examined data from in vivo experiments focusing on the impact of pharmacological interventions or genetic knockouts on proposed signaling pathways determined that the inhibition of nNOS resulted in the most substantial decrease in neurovascular response. Specifically, this blockade led to an average reduction of 64% across 11 studies [24,25]. Lacroix et al. presented compelling evidence that identifies PGE2 as the principal prostaglandin involved in NVC, with pyramidal neurons serving as the predominant cellular source. Their research also highlights the significance of vasodilatory EP2 and EP4 receptors as the main targets of PGE2. These findings were achieved through a multidisciplinary approach that integrated single-cell reverse transcriptase-PCR, mass spectrometry, both ex vivo and in vivo pharmacological methods, and optogenetics [26].

NVC is a dynamic process in which the signaling cascade of the neurotransmitter network and vasoactive mediators can produce corresponding hemodynamic responses to various types of stimuli so that the brain can respond specifically to different types of stimuli [27]. To determine the effectiveness of hemodynamic responses to neuronal activity, more in-depth and comprehensive studies of the physiological and pathological mechanisms of cerebral vessels are needed. However, the neurons that control the dynamics of the cerebral artery are poorly understood. Before assessing the integrity of the neural networks behind vascular changes and deciphering the corresponding excitatory and inhibitory neuronal changes in specific hemodynamic signals, it is crucial to gain a better understanding of these mechanisms.

### 3.2. Glia Cells and NCV

#### 3.2.1. Astrocytes and NVC

Astrocytes, due to their proximity to both synapses and local microvessels, are ideally situated to connect neural activity with microvascular function [28]. Nonetheless, the role of astrocytes and astrocytic calcium (Ca^2+^) in regulating arteriolar tone is still a matter of debate. The conventional view is that neurons release glutamate to activate mGluR5 on astrocytes, causing the release of Ca^2+^ from internal stores, activation of phospholipase A2, and production of vasodilatory arachidonic acid derivatives [29]. Similarly, the induction of neuronal activity through extracellular electrical stimulation elicits Ca^2+^ responses in astrocytes, subsequently resulting in arteriolar vasodilation, a process that is substantially diminished by the application of mGluR antagonists and cyclooxygenase-1 (COX-1) inhibitors [30]. However, further investigations utilizing two-photon laser scanning microscopy (TPLSM) revealed that the expression of mGluR5 in astrocytes is subject to developmental regulation and becomes undetectable beyond three weeks after birth [31]. Thus, the current study is skeptical about the role of astrocytes in functional congestion. An alternative proposed mechanism suggests that Ca^2+^ transients in astrocytes activate large conductance calcium-activated potassium channels located in astrocytic endfeet, contributing to potassium (K^+^) release and subsequent relaxation of VSMCs through inward rectifier potassium (KIR) channels [32]. However, in mice deficient in type 2 IP3 receptors—key regulators of Ca^2+^ release from intracellular stores in astrocytes—NVC occurs despite the absence of astrocytic Ca^2+^ elevation. Conversely, mice lacking a subunit of BK channels demonstrate normal NVC [31]. The expression of calcium signaling in astrocytes remains a controversial topic in the field.

Elevated intracellular calcium levels in astrocytes can lead to the dilation of arterioles, thereby increasing CBF; however, there is a lack of in vivo evidence to support the essential role of astrocytes in functional hyperemia [33]. Institoris et al. found that functional hyperemia consists of two distinct phases, an early and a late component. In awake mice, the dilation of arterioles continues during sustained sensory stimulation. Moreover, they discovered that inhibiting astrocytic Ca^2+^ signaling through the expression of a plasma membrane Ca^2+^ ATPase (CalEx) in vivo reduces the prolonged, but not the transient, sensory-induced dilation of arterioles [34]. Rosenegger et al. previously demonstrated that astrocytes provide tonic regulation of arterioles using resting intracellular Ca^2+^ independent of phasic neuronal-evoked vasodilation [35]. However, astrocyte activity associated with the relaxation of penetrating arterioles or capillaries is controversial. Hatakeyama et al. demonstrated that the activation of astrocytes expressing channelrhodopsin-2 (ChR2) results in the dilation of both pial and penetrating arteries. In contrast, the activation of ChR2-expressing neurons primarily induces the dilation of penetrating arterioles [36]. Furthermore, Mishra et al. established that astrocytes facilitate neurovascular signaling to capillary pericytes but not to arterioles. They proposed that the dilation of arterioles is contingent upon the activation of NMDA receptors and the calcium-dependent generation of NO by interneurons [28].

Indeed, capillary-expressed Kir channels can detect extracellular K^+^ released by astrocytes and neurons, and they facilitate the transmission of hyperpolarization throughout the vascular tree [37]. In vivo, capillary dilation precedes arteriolar dilation in response to increased sensory stimulation, contributing to an estimated 84% increase in CBF [38]. Low-frequency stimulation consistently produces rapid neuronal Ca^2+^ transients but does not always cause astrocyte Ca^2+^ transients [39]. However, high-frequency stimulation causes considerable delay in astrocyte Ca^2+^ accumulation and is associated with transient vasoconstriction [39]. When the astrocyte transient occurred, the intensity of hemodynamic response increased. These suggest that Ca^2+^ transients in astrocytes in response to neuronal activity are sparse and delayed relative to the onset of functional congestion [39].

#### 3.2.2. Microglia and NVC

Previous studies on microglia and neural vascular coupling are limited; however, recent studies have identified specific locations on neuronal cell bodies that facilitate the modulation of neuronal responses by microglia through purinergic mechanisms [40]. sászár’s team found that microglia modulated blood flow, NVC, and hypoperfusion via purinergic actions [41]. LSM was used to examine the dynamics of microglia-neurovascular interactions. Microglia frequently contact the same sites on blood vessels, with a mean contact lifetime of 5–15 min. P2Y12 receptor (P2Y12R)-positive microglia are in contact with 85% of the vascular segments, 83% of the pericytes, and 93% of the astrocytes [42]. The lack of microglia or the inhibition of the microglial P2Y12R significantly disrupts NVC in mice, a finding supported by the observation of chemogenetically induced microglial dysfunction linked to blueuced sensitivity to ATP [42]. These findings indicate that purines released via pannexin 1 (PANX1) channels are crucial for the activation of microglial P2RY12 receptors, which, in turn, regulate neurovascular structure and function.

Perivascular microglial processes exhibit a swift response to hypercapnia, characterized by calcium pulses and the formation of new filopodia, alongside the rapid production of adenosine by microglia. Recent studies indicate that microglia serve as a crucial source of adenosine in the brain, playing a significant role in modulating neuronal responses at synaptic sites [43]. Additional research has supported the concept of purinergic signaling between glial and ECs during neural activity, emphasizing the importance of ATP in triggering eNOS activation through P2Y1 receptors, which is crucial for NVC [44]. Microglia are in dynamic contact with NVU, and the specific role of microglia in NVC is still in the preliminary stage. Astrocytes have a significant role in NVU, sensing neuronal activity through their peri-synaptic processes and releasing transmitters at the end of contact with blood vessels in the brain parenchyma. Although there are several controversies in this field, with continuous innovation in experimental technology, our understanding of the regulatory role of glial cells will improve.

### 3.3. Endothelial Cell and NCV

The role of microvascular ECs in NVC has often been underestimated. ECs are known to influence the retrograde propagation of vasomotor responses within the context of NVC [44]. Despite this knowledge, the precise mechanisms that govern retrograde vasodilation in NVC are still not fully elucidated. Recent studies, however, indicate that cerebrovascular ECs may possess the capability to detect norepinephrine (NA) and synaptic activation, triggering a hemodynamic response that facilitates the propagation of vasodilation to upstream vessels and surface arteries [45,46]. The signal transduction phenomenon within ECs during NVC primarily involves endothelial inward-rectifier potassium (K_ir_2.1) channels and the N-methyl-D-aspartate receptor (NMDAR)-mediated release of eNO [45,47,48]. The activation of Kir2.1 leads to hyperpolarization of the endothelium, a change that can be electrotonically conveyed to adjacent pericytes through myoendothelial gap junctions, ultimately initiating vasodilation by decreasing intracellular calcium levels [49].

Thakore et al. found that depletion of phosphatidylinositol 4,5-bisphosphate (PIP2) in ECs leads to the loss of Kir2.1 channel activity in aged animals, while phosphatidylinositol-3-kinase (PI3K) inhibition restored the Kir2.1 channel, NVC, and cognitive impairment associated with CSVD [50]. Anfray et al. showed that circulating tissue plasminogen activator (tPA) could enhance the increases in CBF triggered by the activation of the mouse barrel cortex [51]. The tPA-dependent activation of NMDAR plays a key role in mediating the effects observed in the ECs of arteries and arterioles [51]. Research has shown that endothelial NMDARs play a crucial role in facilitating glutamate-induced vasodilation in intraparenchymal arterioles. This process is achieved through the mediation of Ca^2+^ influx, which subsequently activates eNOS [22,52]. Moreover, NMDAR-induced NO signaling targets connexins to favor the propagation of endothelial hyperpolarization to the (feeding arteriole) vessels located upstream, rather than downstream, of the site of neuronal activation [53].

The debate regarding the primary origin of NVC—whether it arises from the capillaries or the arterioles—has been enriched by several recent studies that compellingly demonstrate that capillary relaxation may precede arteriolar dilation. This sequence is attributed to NA-induced relaxation of the contracting pericytes that encase the capillary walls [37,54,55]. Notably, transient receptor potential ankyrin 1 (TRPA1) channels present in the ECs of capillaries are activated by neuronal activity, leading to the generation of a propagating retrograde signal. This signal subsequently induces dilation of the upstream parenchymal arterioles, thus initiating functional hyperemia [50]. Specifically, calcium (Ca^2+^) signals that originate from TRPA1 channel activation gradually propagate through the capillary network through a previously unidentified pathway. This pathway is contingent upon the activity of ionotropic purinergic P2XR and the expression of PANX1 [50]. Furthermore, PIP2—a minor phospholipid located in the inner leaflet of the plasma membrane—might function as a molecular switch that modulates the efficiency of electrical signaling in brain capillaries. It also regulates the upstream propagation of hyperpolarization mediated by ECs [56].

PIP2’s role extends to the modulation of the Kir2.1 channel in brain capillary ECs, which fosters hyperpolarization, as well as the transient receptor potential vanilloid 4 (TRPV4) channel, which exhibits permeability to sodium (Na^+^) and calcium (Ca^2+^) ions, thereby promoting membrane depolarization [57]. In contrast to other vascular segments within the central nervous system (CNS), arteriolar ECs are characterized by a high density of caveolae [58]. Acute genetic alterations leading to the depletion of caveolae in arteriolar endothelial cells (aECs), while sparing neighboring VSMCs, have been shown to disrupt NVC. Significantly, the caveolae in aECs function independently of the eNOS-dependent NO signaling pathway [58]. The complete ablation of both caveolae and eNOS resulted in a total loss of NVC, while single mutants displayed partial deficiencies. This illustrates that the caveolae-dependent pathway in aECs is a crucial component of NVC, critically facilitating the transmission of signals from the CNS to VSMCs through a caveolae-mediated mechanism [58,59]. Various ion channels and special structures in ECs are key regulatory factors in NVC signal transduction [60]. Further elucidation of how EC interacts with other components of the NVU, the hemodynamic response of different vascular segments, and the differences between different brain regions is required.

### 3.4. SMCs, Pericytes, and NVC

Signals generated by neurons, astrocytes, and ECs ultimately act on VSMCs and pericytes to induce vasodilatation, reduce vascular resistance, and increase blood flow [55]. The vasoconstrictor and vasodilator apparatus at the arterial and arteriolar levels are predominantly VSMCs [23]. Pericytes are found on precapillary arterioles, capillaries, and post-capillary venules, which extend the primary processes along the abluminal and longitudinal axes of blood vessels [24,61]. It has been difficult to distinguish pericytes from VSMCs at the arteriolar–capillary transition because of the lack of specific markers. The smooth muscle that perforates arterioles is rich in α-smooth muscle actin (α-SMA) and has a strong contractile function [62]. It also has a layer of elastin that can be labeled with Alexa633 hydrazide [63]. As microvessels continue to branch, there is a significant decrease in the expression of α-SMA, dropping to low or undetectable levels. This change delineates the transition from the arteriole-capillary junction to the true “capillary” zone. It is characterized by a shift from coverage by ensheathing pericytes to the presence of capillary pericytes exhibiting classic pericyte morphology [61].

Initial studies indicated that regional blood flow in normal and ischemic brains is controlled by arteriolar SMCs contractility and not by capillary pericytes [64]. Likewise, it was shown that precapillary and penetrating arterioles, rather than pericytes in capillaries, are responsible for functional hyperemia induced by neural activity [65]. However, an increasing number of studies have confirmed that capillary pericytes play a critical role as key effectors of neuronal- and glial-derived signals in the regulation of cerebral microvascular blood flow [54]. Precapillary sphincters and pericytes located on the initial branches of capillaries may elicit more pronounced and rapid changes in vascular diameter compared to VSMCs found in arteries and arterioles [66]. Further investigations into the aging mouse brain using optogenetic models and TPLSM revealed that pericytes are significant regulators of capillary blood flow [67]. Mathematical modeling based on 3D vascular reconstructions has demonstrated that precapillary sphincters and first-order capillary pericytes act as crucial sensors and effectors of vascular signals derived from the endothelium or brain [68]. Kisler et al. demonstrated that mice lacking pericytes exhibit diminished capillary CBF responses to induced stimuli, resulting in a decreased oxygen supply and heightened metabolic stress [7]. Recently, it has been further demonstrated that the acute loss of pericyte coverage rather than other vascular deficits or neuronal dysfunction serves as the primary driver of the observed deficits in NVC [69]. Nikolakopoulou et al. showed that inducible pericyte knockdown in adult mice results in basal CBF deficiency and disruption of the BBB [70]. Bisht et al. identified P2RY12 receptors as regulators of capillary-associated microglia (CAMs) interactions under the control of purines released from PANX1 channels at the molecular level [71]. Their results showed that the excitation of the photosensitive ChR2 led to pericyte contraction, followed by capillary constriction and a significant decrease in red blood cell flow [71]. The optical ablation of individual capillary pericytes resulted in prolonged local dilation and a doubling increase in blood cell flux specifically within capillaries that were devoid of pericyte contact. This observation suggests that capillary pericytes play a crucial role in regulating basal blood flow resistance and modulation across the brain [72].

Despite these advancements, the in vivo signals that lead to altered vascular dynamics remain unclear. Differences in the details of excitation–contraction coupling between pericytes and VSMCs, including differences in Ca^2+^ transients, indicate that the mechanism is complex. The complex excitation–contraction coupling in VSMCs has been well-documented in recent years. These studies also confirmed that pericytes can convert stimuli into contractile responses through excitation–contraction coupling. However, this coupling mechanism is inconsistent in parietal cells with different vascular segment phenotypes, which may require selective intervention. Novel imaging techniques such as TPLSM imaging of living animals, optogenetics, and calcium kinetic have provided new insights into the cellular elements involved in NVC.

### 3.5. Cellular Interaction and NVC

Currently, NVC-related research focuses on two major modules: one is to conduct in-depth research on the mechanism of a cell’s role in the NVC (Table 1); the other is to explore whether the phenomenon of neurovascular uncoupling occurs after the onset of disease, and the potential mechanism and whether this phenomenon affects neurological function (Table 2). The cellular components of the NVU may lead to neurovascular uncoupling through complex interactions. Local CBF is directly affected when neuronal activity is reduced or abnormal. Neurons affect the peripheral vasculature by releasing metabolites and neurotransmitters such as glutamate, whereas astrocytes and microglia are involved in regulating vascular responses by releasing vasoactive substances such as prostaglandins and NO [22,35]. In pathological states, changes in the reactivity of these cells may lead to vasodilatory dysfunction, which, in turn, affects CBF. The altered activation state of astrocytes may lead to an imbalance in their release of vasoactive substances, whereas overactivation of microglia may exacerbate inflammatory responses and vascular injury [41]. Dysfunction of VSMCs and ECs may lead to reduced vascular reactivity and BBB disruption, further affecting NVC [73,74]. Abnormal contraction of pericytes at the capillary level may lead to increased vascular permeability, exacerbating brain edema and inflammation [38]. Together, these factors lead to altered vascular reactivity, disruption of the BBB, hemodynamic disturbances, and increased neuroinflammation, ultimately leading to uncoupling of NVC and affecting brain tissue repair and functional recovery. However, these are just some basic theories, which do not have much guiding value for clinical translation. Future research should focus on exploring how much damage to neurons or other cellular components causes neurovascular uncoupling, and how much neurovascular uncoupling causes neurological dysfunction. This will guide us in the search for targets for intervention and the application of therapeutic measures.

### 3.6. Collateral Circulation and NVC

The role of cerebral collateral circulation in NVC is a complex and finely regulated process that involves a variety of factors and mechanisms to ensure that, in the event of stenosis or occlusion of the main supplying arteries, blood flow is provided to the ischemic region via other vascular pathways to maintain perfusion and function of the brain tissue. In NVC, the role of cerebral collateral circulation is not limited to providing alternative blood flow pathways. It is also involved in the fine-tuning of neural activity [107].

Binder NF et al. revealed the role of Leptomeningeal collaterals (LMCs) in stroke reperfusion [108]. They investigated this by laser scatter imaging, ultrafast ultrasound, and two-photon microscopy techniques in a thrombin-based mouse stroke model and found that LMCs maintained autoregulation of CBF and allowed gradual reperfusion, resulting in smaller infarct size [108]. In mice with poorly functioning LMCs, distal arterial segments collapsed, and deleterious high blood flow after reperfusion led to hemorrhage and death. These findings suggest that LMCs are a key component in regulating reperfusion and preventing useless recanalization after stroke [108]. The study by Ravindran et al. looked at the effects of cerebral collateral circulation in patients with acute ischemia undergoing reperfusion therapy [109]. They emphasized the importance of cerebral side-branch circulation in stroke patients undergoing reperfusion therapy, particularly in predicting treatment outcomes [109]. The results of these studies suggest that the state of the cerebral side branch circulation has a significant impact on successful revascularization and treatment outcomes.

The “Evolucollateral Dynamics” hypothesis proposed by Sinha et al. [107]. presents a novel framework for incorporating evolutionary biology principles into therapeutic strategies, providing new perspectives for enhancing collateral circulation. These studies suggest that the development and function of LMCs may be regulated by genetic predisposition and environmental influences, which have important implications for the development of therapeutic strategies and optimization of treatment outcomes. This hypothesis highlights the role of genetic predisposition and environmental influences on collateral circulation, which may influence treatment strategies and optimize therapeutic outcomes. Future studies must incorporate human clinical data to create robust therapeutic regimens that maximize the therapeutic potential of LMCs and improve the prognosis of stroke patients.

## 4. Neurovascular Coupling: Impairments and Diagnostic Assessments in Cerebrovascular and Neurological Disorders

Having detailed the cellular mechanisms of NVC, we have gained a deep understanding of the molecular and cellular level changes involved in this complex physiological process. This basic science knowledge has shed light on the central role of NVC in maintaining brain function and provided us with a starting point for exploring its changes in disease states. A fundamental factor in impaired NVC is insufficient cerebral perfusion pressure, which may cause a decrease in cerebrovascular tone secondary to futile neuronal activation, resulting in congestion disorders [110]. In contrast, congestive signaling is altered by glutamate release secondary to neuronal death and astrocyte scarring [111]. Consequently, NVC impairment may occur in various CNS diseases such as cerebrovascular disease, Parkinson’s disease, traumatic brain injury, and cognitive impairment. Studies have confirmed that oxidative stress has a significant destructive effect on NVC and is associated with cerebrovascular disease, hypertension (angiotensin receptors), Alzheimer’s disease (AD), and SAH, and is also observed in healthy aging individuals [112,113,114,115,116].

Current advances in imaging techniques for nerve cells and CBF allow us to observe, manipulate, and understand the cellular activity of members of the NVU in greater depth. We can observe how NVC affects overall brain function and disease progression by studying NVC in disease through these imaging techniques. Functional magnetic resonance imaging (fMRI), based on BOLD, uses the difference between oxygenated and deoxygenated hemoglobin in response to a magnetic field to form different signals that indirectly reflect the functional activity of neurons [117]. It can objectively reflect changes in the activity of various functional areas of the brain in real time. Arterial spin labeling (ASL) refers to arterial spin labeling of blood flowing into the brain through a radiofrequency pulse [118]. Based on the signal difference in blood, information on cerebral blood perfusion is obtained and can be calculated from the quantitative CBF data. High temporal resolution electroencephalography (EEG) and high spatial resolution functional near-infrared imaging (fNIRS) complement each other and facilitate a more comprehensive study of NVC mechanisms in the brain during working memory [98]. These techniques allow NVC to be evaluated in both time and space, and it is already being used in the clinic (Figure 3).

However, these techniques either do not allow high resolution monitoring of blood flow below the surface of the cortex or suffer from low spatiotemporal resolution. In addition, none of these techniques allow imaging of blood flow and cell activity at the same time. The TPLSM technique can observe both blood flow and cell activity at the same time, while traditional TPLSM cannot observe vasculature and cell imaging deeper beneath the cortex [119]. The new TPLSM module provides many advanced features for NVC research. By using longer excitation wavelengths, and because the light absorption used to excite fluorescent molecules is largely confined to the focus, TPLSM has a key improvement in its ability to image tissues at greater depths with less phototoxicity [120]. However, the high economic cost of TPLSM technology makes its popularization limited. In addition, TPLSM requires acute or chronic cranial Windows or tissue transparency techniques, which limits its clinical translation. In summary, imaging technologies provide powerful tools for NVC research, but technical challenges and limitations still need to be overcome to achieve a more comprehensive understanding of NVC. Future research needs to further optimize these techniques to improve imaging depth, resolution, and temporal resolution while reducing cost and technical thresholds. We display how NVC is induced in various clinical diseases, assessment tools, etc. in Table 2. We will now move from the cellular level to a more macroscopic view of NVC research in specific diseases. This will not only help us understand the specific manifestations of NVC in pathological states but will also be crucial for developing therapeutic strategies for specific diseases.

### 4.1. Ischemic Stroke

Over the past few decades, animal models of stroke have been widely utilized to elucidate the mechanisms underlying stroke recovery. These studies have provided insights ranging from cellular and molecular alterations to the reorganization of functional networks [121,122]. However, limited work has been conducted on the integration of neural and hemodynamic measures to study neurovascular recovery after stroke. Acute ischemic stroke produces excitotoxicity and diffuse depolarization (SD) owing to the excessive release of glutamate, leading to the constriction of small arteries [52]. SD is caused by local ischemia of brain tissue, where the ionic balance of neurons and glial cells is disrupted, leading to extensive depolarization of cell membrane potentials. Contractile VSMCs control vasomotor activity depending on cytoplasmic Ca^2+^ oscillations. Excess glutamate may disrupt VSMCs Ca^2+^ homeostasis, which may be related to VSMCs contraction behavior during SD, thus expanding the infarct size [4,123]. Baker et al. found that NVC varied with the level of global cerebral ischemia in a rat model [124]. He et al. integrated flexible intracortical electrode arrays with laser speckle contrast imaging (LSCI) to concurrently assess LFP, spiking activity, and CBF throughout the process of chronic stroke recovery in the somatosensory cortex [125]. They observed significant neurovascular disassociation immediately after the stroke and in the subacute phase. Furthermore, the severity of neurovascular disassociation in the chronic phase is associated with the degree of initial injury [126]. This study proposes that vasoconstrictive NVC during ischemic depolarization plays a significant role in the hemodynamic progression associated with acute focal cerebral ischemia. Furthermore, it suggests that alleviating the detrimental vascular effects resulting from tissue depolarization could be a vital mechanism through which neuroprotective drugs mitigate tissue damage [127,128,129]. Sunil et al. proved that NVC is preserved in the chronic phase of recovery and resembles the pre-stroke state in the photothrombotic stroke model [130]. Povlsen et al. found that astrocytic Ca^2+^ signaling remained normal, suggesting a possible defect in parenchymal VSMCs that respond to released vasodilator substances caused by impaired Kir channel function [127]. Li et al. found that cortical 20-HETE levels were increased following middle cerebral artery occlusion (MCAO), and the inhibition of ω-hydroxylases normalized 20-HETE levels in vivo and increased CBF in the peri-infarct cortex [18]. This study suggests that 20-HETE-dependent vasoconstriction may be a mechanism underlying capillary NVC impairment after stroke.

After ischemia and reperfusion, transient hyperperfusion is followed by secondary hypoperfusion, and the area of hypoperfusion gradually expands with the increase in metabolic demand around the infarction [131]. Pericapillary cell contraction and capillary flow arrest in the ischemic core. Staehr et al. showed that capillary dysfunction was related to neurovascular decoupling in a mouse model of photothrombotic ischemic stroke [132]. Peroxynitrite-mediated pericellular contraction in ischemic lesions may hinder the perfusion of the capillary bed, even when the proximal artery has been reperfused [132]. These phenomena may explain futile reperfusion after ischemia/reperfusion [133]. Several clinical studies have confirmed that NVC is impaired in patients with ischemic stroke. Wu et al. analyzed the causal relationship between CBF and neuronal activity in stroke patients using transcranial Doppler sonography (TCD) and EEG, and the intensity of this relationship was related to stroke severity [84]. Salient et al. conducted a similar study and found that brain autoregulation in stroke patients was impaired and correlated with stroke severity compared to normal individuals [87]. The authors suggested that the loss of CBF in response to neuronal activation after stroke may be due to myogenic damage. Huneau et al. found that dysfunction of NVC occurred in the early stages of cerebral autosomal dominant arteriopathy with subcortical infarcts and leukoencephalopathy (CADASIL) based on fMRI and EEG [88]. Wang et al. analyzed the alterations in resting-state NVC in patients with pontine infarctions [5]. The results indicated that CBF-FCS coupling was significantly interrupted in patients with primary immunodeficiency (PI), and changes in NVC were correlated with working memory scores [5]. It is important to observe the dynamic neurovascular responses at the onset of ischemia and after reperfusion. Further studies linking the hyperacute regulation of NVC after ischemic stroke to long-term tissue, neurovascular imaging, and nerve function will facilitate the development of new hyperacute interventions that can help preserve cerebrovascular autoregulation after stroke and prevent reperfusion injury.

### 4.2. Subarachnoid Hemorrhage

SAH induces acute changes in cerebral microcirculation and neurovascular uncoupling. Recent ex vivo findings have suggested that NVC is also impaired after SAH [134]. For instance, CO2 reactivity was completely lost in both the pial and parenchymal arterioles, whereas the NVC was unaffected [3]. Both processes were massively impaired when CO2-reactivity and NVC were investigated 24 h after SAH [3]. However, the pial and parenchymal vessels were dilated in response to CO2, and NVC was almost absent 1 month after SAH [3]. Therefore, microvascular dysfunction persists over the long term and may lead to a long-lasting mismatch between neuronal activity and CBF. Koide et al. found an inversion of NVC after SAH in vivo, with constricted rather than dilated arterioles in response to neuronal activity 24–96 h after SAH [135]. Balbi et al. further proposed that the impairment of NVC after SAH occurs secondarily and is progressive [136]. Neuronal activity-induced vasoconstriction (inverse NVC) is likely to aggravate SAH-induced cerebral ischemia and subsequent brain damage [136]. Pappas et al. also reached the same conclusion, and their research indicated that purinergic signaling via P2Y receptors may contribute to the SAH-induced high-affinity Ca^2+^ influx system and inversion of NVC [134]. Furthermore, the pathophysiology is primarily attributed to increased basal levels of perivascular K^+^, which result from the heightened amplitude of spontaneous Ca^2+^ oscillations and the augmented activity of large-conductance BK channels in the endfeet of astrocytes [135].

Conzen et al. studied the retinal arterioles of patients with SAH 5 days after the onset of the disease [137]. Using retinal angiography, although they did not find a reversal of NVC, they observed that the arterioles in the acute stage of SAH decreased in diastolic diameter after receiving stimulation [137]. Moreover, the time required to reach the maximum diastolic diameter increased, although these conditions partially improved after 3 months [137]. However, there was still a difference between the control and SAH groups, which indicated that NVC impairment was present in patients with SAH and that this impairment was not recoverable in the short term [137].

Further investigation using retinal vascular analysis (RVA) explored the association between delayed cerebral ischemia (DCI) and NVC disorders. At the early stage of SAH (0–4 days), retinal arteries in the DCI group took less time to reach 30% of the maximum diastolic diameter upon stimulation compared to the non-DCI group. Conversely, at the late stage of SAH (16–23 days), the DCI group took longer to reach this milestone than the non-DCI group [106]. These subtle differences suggest potential relationships between NVC reversals and DCI, warranting further investigation in larger trials. Pathological studies on NVC disorders in SAH indicate that addressing NVC reversal early may help restore CBF and reduce the incidence of delayed cerebral ischemia. While this hypothesis requires more experimental validation, these findings emphasize the importance of early intervention in the management of SAH-induced NVC disorders.

### 4.3. Cerebral Small Vascular Disease

Studies have shown that CSVD plays a key role in the automatic regulation of CBF and that changes in cerebral perfusion caused by arteriosclerosis are one of the main factors of CSVD, suggesting a close relationship between CSVD and NVC [138]. Damage to the cerebral microvascular injury may diminish vasodilation reserves and impair NVC function, leading to a decline in cognitive function and reduced CBF reactivity [5]. A systematic review examined the relationship between CSVD and NVC across 25 clinical studies [139]. The findings revealed that NVC damage in CSVD is associated with imaging markers such as white matter lesions, cerebral microbleeds, and lacunar strokes [139]. Notably, no studies reported a relationship with perivascular space enlargement. The degree of NVC damage was found to increase with the severity of white matter lesions and microbleeds.

Neurovascular dysfunction has also been observed in animal models of CSVD. A series of pathological changes in CSVD, including damage to the cerebral microvascular ECs, enhanced the permeability of the BBB, enhanced the oxidative stress of the microvascular system, impaired NOS/NO signaling pathway, and impaired NVC [140]. Wiedenhoeft et al. found that a fusion liposome based on resveratrol could directly target cerebral microvascular ECs; enhance NO-mediated vasodilation; and significantly improve the NVC response, microvascular function, and cognitive function [141]. In a study focusing on CADASIL patients, selective vasomotor dysfunction has been identified [86]. Furthermore, researchers have used a model of direct co-culture between neurons and microvascular ECs to study the interaction between neurons and ECs, and the results showed that cerebral microvascular ECs can regulate the ion channel activity of neurons and alleviate hypoxia and ischemic damage in neurons [142]. Most previous studies have focused on the protection of single neurons or blood vessels. However, the concept of NVC has shifted the focus toward global neurovascular protection, becoming a hot spot in CSVD treatment approaches. Thakore et al. report that the Col4a1 mutant mouse model of CSVD exhibited age-dependent deficits in capillary-to-arteriolar dilation, functional brain congestion, and memory [82]. This defect was related to the depletion of the minor membrane PIP2 in brain capillary ECs, resulting in the loss of activity of the Kir2.1 channel. Reducing PIP2 consumption by blocking PI3K restored Kir2.1 channel activity, followed by capillary–arteriole dilation and functional congestion [82]. This suggests that PI3K inhibition is a potentially viable therapeutic strategy for treating deficient NVC and cognitive impairment associated with CSVD. Therefore, promoting NVC remodeling to improve the microcirculation of small cerebral vessels may be an effective measure to counteract a series of neurovascular injuries caused by CSVD and even block the development of CSVD and cognitive dysfunction.

### 4.4. Other CNS Diseases

Alzheimer’s disease (AD) is characterized by progressive memory decline and deficits [143]. It has been widely reported that NVC, a vasodilatory response to neuronal activity, is impaired in patients with AD or animal models [47]. Reactive oxygen species (ROS) are elevated in astrocytes of amyloid-beta precursor protein (APP) mice, and Aβ-induced ROS is known to alter vascular regulation [144,145,146,147]. Shabir et al. found that Ca^2+^ uncaging-induced ROS production was elevated in APP mice, and ROS scavenging rescued hippocampal NVC impairment while normalizing astrocytic Ca^2+^ signaling in APP transgenic mice [148]. Bonnar et al. studied the visual cortex and hippocampus of awake apolipoprotein E (APOE3) and APOE4 target replacement (TR) mice using TPLSM of NVU [149]. They found that vascular structure and functional hyperemia were unaffected in APOE4 mice [149]. Instead, vascular responsiveness was lower, arteriolar vasomotion was reduced, and neuronal calcium signals increased during visual stimulation. APOE4 expression was not catastrophic but more sensitive to subsequent insults such as injury or Aβ accumulation [150,151]. Bonnar et al. compared the efficacy of NVC in the cerebral cortices of transgenic Alzheimer’s (CVN-AD) and control (C57BL/6) mice [149]. The results suggested that CVN-AD Alzheimer’s mice had a premature reduction in NVC in response to increasing depression and anoxia compared to aged controls. Therefore, NVC coupling disorders may precede the pathological development of AD. AD is not only the degeneration of neurons at the early stage; the disorder of CBF regulation accelerates AD pathology development, and the deposition of Aβ damages blood vessels [16]. Therefore, the understanding of mechanisms underlying changes in NVC during the early stages of AD is important.

Considering other CNS diseases, Hu et al. studied neuronal activity and cerebral hemodynamic changes in patients with idiopathic generalized epilepsy (IGE) by calculating the degree centrality (DC) and CBF using fMRI and ASL [96]. The results suggested a higher CBF/DC ratio in the right posterior cingulate cortex/precuneus, middle frontal gyrus, and medial frontal gyrus (MFG) and a lower ratio in the left inferior frontal gyrus in those with IGE than that in healthy controls (HCs) [96]. The increased CBF/DC ratio in the right MFG correlated with lower performance intelligence quotient scores in the IGE group. Researchers have also found impairment of NVC in patients with headaches and multiple sclerosis who were overmedicated [100,101]. Clinical studies have demonstrated impaired NVC in tinnitus, sleep disorders, and aging individuals [102,152,153]. Toth et al. proved that age-related circulating insulin-like growth factor-1 (IGF-1) deficiency contributes to neurovascular aging and impairs CBF and NVC in older [89]. Csipo et al. found that 24 h of sleep deprivation leads to impairments in cognitive performance, along with altered CBF and hemodynamic components of cortical NVC responses [89]. Subcortical ischemic vascular disease (SIVD) with cognitive impairment showed a more severe decoupling of the global ReHo-CBF correlation based on fMRI and ASL, which indicated severe disorders of NVC [91]. Their study showed that patients with SIVD had abnormal NVC at the early stage of the disease and during its development. The neuropathological mechanism of SIVD may be associated with disturbances in NVC. NVC has the characteristics of multi-structure participation and multi-pathway regulation and is closely related to the maintenance of brain homeostasis. These studies suggest that NVC dysfunction occurs in a variety of diseases even at an early stage and that NVC disorders may play a key role in the occurrence and development of the disease. This phenomenon is more common in older than in younger individuals. Early clinical attention and evaluation of NVC function may be of great help in the early recognition of the disease.

## 5. Potential Intervention Targets and Strategies

To identify potential intervention targets and strategies, recent research has explored various approaches to mitigate neurovascular dysfunction. A study treated 24-month-old C57BL/6 mice with NMN, a NAD intermediate, finding that NMN supplementation improved NVC responses by raising eNO-mediated vasodilation, resulting in promoting spatial working memory and gait coordination [154]. This suggests that reduced NAD availability related to age-related cerebromicrovascular dysfunction and cognitive impairment, making NAD intermediates potential interventions for vascular cognitive impairment. Supporting this, PJ-34 and SS-31 were shown to improve cerebromicrovascular ECs function and cognitive in aged mice [155,156]. mTOR, which drives cerebrovascular dysfunction in AD models by reducing eNOS activity, can be targeted with rapamycin to restore NVC function [157,158]. Oxidative stress-induced NVC dysfunction, implicated in VaD, was mitigated by LG, which may protect NO bioavailability and maintain NVC sensitivity [159]. Additionally, tPA in circulation was found to influence NVC by potentiating CBF increase through a tPA-dependent activation of NMDAR on ECs [51]. According to the latest research findings about the mechanism of NVC, some potential future intervention targets and innovative intervention approaches have also emerged. Research by Jiemin Jia’s team at Westlake University has revealed the mechanism of direct dialog between neurons and cerebral blood vessels, a finding that provides new therapeutic ideas for NVC dysfunction [4]. They found that there are “holes” in the wrapping rate of astrocyte end-feet to the peripheral arteries, and the presynaptic sub-synapses of perivascular neuron axons will cross these holes and form “synapse-like” structures with VSMCs, which provide a new bridge for the dialog between neurons and blood vessels [4]. Using optogenetic activation techniques, the team demonstrated that this structure can regulate vasoconstriction and diastole in the brain, thereby affecting NVC [4].

The NsMJ structure allows glutamatergic neurons to act directly on arterial VSMCs, leading to arteriolar diastole and inducing functional cerebral congestion [4]. This finding provides new insights into understanding the rapid and precise regulation of CBF and offers new ideas for the effective treatment of ischemic hypoperfusion injury. The team also proposed a comprehensive and systematic vascular dynamics analysis algorithm, which not only describes the basic characteristics of cerebral arterial vasomotion in mice under physiological conditions but also records and analyzes in real time the damage and recovery of cerebral arterial vasomotion and CBF encounters in ischemic stroke conditions in mice [160]. Reactivation of spontaneous vasomotion in cerebral arteries by transgenic (ME-Linker) mice achieves optimal reperfusion while ameliorating neurological damage and contributing to cerebrovascular recovery after ischemic stroke [160]. These research advances suggest that new therapeutic approaches can be developed through an in-depth understanding of the molecular and cellular mechanisms of NVC, thereby improving the therapeutic outcomes of neurological diseases such as stroke. These results not only deepen our understanding of the role of NVC in the regulation of CBF but also provide a scientific basis and a new direction for future clinical treatment.

In addition, research by Yan Dou’s group at Tianjin Medical University General Hospital has proposed a cross-scale targeted nanotherapy based on the natural self-assembling SIRT1 activator Ren [161]. This approach achieves precise and efficient treatment of TPLSM by specifically targeting damaged cerebral microvascular ECs, utilizing a short peptide of the receptor for advanced glycosylation end products (RAP) coupled with an amphiphilic block copolymer, PLGA-PEG, and further self-assembled with ellagic acid [161]. NO releases and improves the integrity of the BBB, while promoting neuronal mitochondrial biogenesis and the conversion of glucose metabolism to oxidative phosphorylation, thus remodeling the neurovascular and neurometabolic coupling [161]. Their study reveals a direct dialog mechanism between neurons and cerebral blood vessels, a finding that is not only theoretically important but also shows potential clinical applications.

The potential value of these innovative approaches lies in the fact that they not only provide a new understanding of dysfunctional NVC but also provide a scientific basis for the development of new therapeutic approaches. The future treatment direction of NVC is gradually changing from the single neuroprotection to the whole neurovascular unit protection. This shift is based on a deeper understanding of NVC, especially given the complex interactions between blood vessels and nerves. In the future, the exploration of multi-target brain cell protective agents will become one of the important research and development directions for the treatment of AIS, which means that drug therapy will protect the whole neurovascular unit from different angles and multiple ways. Exercise, rehabilitation therapy, and transcranial electrical stimulation have also been shown to modulate neural activity and CBF between different brain regions, suggesting that non-invasive neuromodulation may also be an important direction for future treatments [162]. However, intervention methods for these potential targets remain largely experimental and have not yet been validated in clinical trials. In the future, we need to further evaluate the feasibility, safety and efficacy of these interventions in clinical trials.

## 6. Advancements and Challenges in NVC

There are several controversies and ambiguities in current NVC research, and these are mainly centered on the regulatory mechanisms, influencing factors, and relationship with disease in NVC. First, in terms of the regulatory mechanisms of NVC, although neuronal activity can influence CBF through multiple pathways, the specific molecular and cellular mechanisms remain incompletely understood. For example, the roles of astrocytes and microglia in NVC are controversial, and they may regulate CBF by releasing vasoactive substances or participating in inflammatory responses, but the specific mechanisms of action and the extent of the effects are yet to be further investigated. Secondly, the interaction between NVC and the BBB is also a hot research topic, and the disruption of the BBB may affect NVC. Still, the specific roles and regulatory mechanisms of the BBB in NVC remain unclear. In addition, the changes in NVC in different brain regions and disease states are controversial. For example, in diseases such as AD and depression, it is inconclusive whether the changes in NVC are the cause, the result, or both. At the technical level, in vivo imaging techniques can provide information on the dynamics of NVC. The development of these more advanced technologies may bring us a new dawn. While in vitro experiments can provide detailed information at the molecular level, how to correlate this information with in vivo status remains a challenge. Finally, the individual variability of NVC is also an area of ambiguity; differences in NVC between individuals may influence disease onset and progression, but the understanding and control of these differences are currently limited. In addition, each component of the NVU has a vital role in the transmission of information. What are the contributions of neurons, glial cells, and vascular recoupling? Which of the perforating arteries, precapillary arteries, or capillaries is the leading segment that regulates cerebral microvascular blood flow? These are not established in the current research. The cellular communication involved and the underlying molecular mechanisms are complex. Large-scale omics, imaging markers, and novel molecular biomarkers are immediately required for further mechanistic exploration.

In addition to the examples discussed here, there is emerging evidence for the functional regulation of NVC throughout life activities. For example, sleep and circadian mechanisms appear to regulate BBB permeability, at least in invertebrate models [163]. It would be interesting to determine if and how these findings apply to mammals. However, it is not clear whether neural activity causes vasomotor activity and vice versa or whether there is only a correlation. Sensitive real-time monitoring of vasoconstriction and relaxation, as well as real-time stimulation and imaging of neural activity, may help elucidate this relationship. In addition, the growing toolkit of optogenetics, chemogenetics, and biosensors provides a means for controlling and detecting the activities of living animals. These methodologies have the potential to address several critical questions in the field of NVC. The primary challenge moving forward is to develop therapeutic strategies to address neurovascular dysfunction by targeting impaired NVC, thereby preventing disease progression. Changes in NVC may vary over time, especially in neurodegenerative diseases. It is also a challenge to develop techniques that allow the stable monitoring of NVC over time.

## Figures and Tables

**Figure 1 cells-14-00016-f001:**
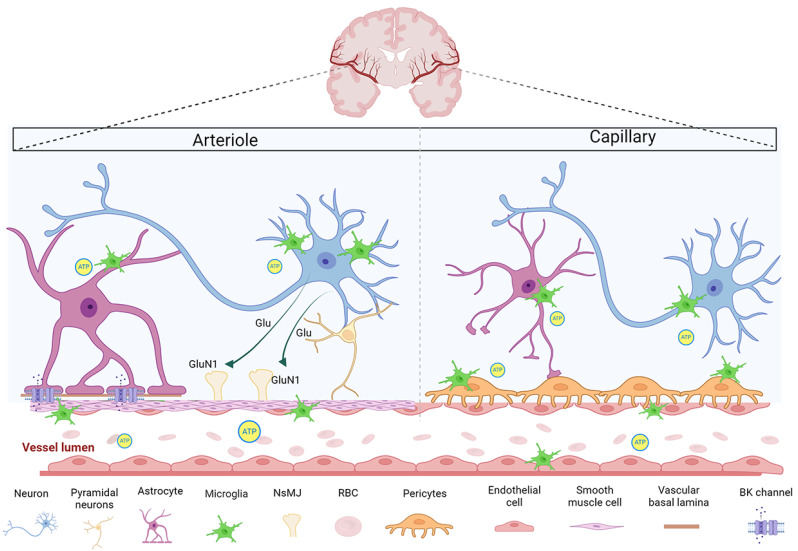
Important component in neurovascular coupling. These include neurons, astrocytes, microglia, endothelial cells (ECs), smooth muscle cells (SMCs), and pericytes. Perivascular endfeet formed by multiple astrocytes surround ECs and allow complex intercellular communication to help maintain the blood–brain barrier (BBB) phenotype of brain ECs. The presence of BK channels in astrocyte foot processes mediates NVC. Microglia aggregates under the chemotaxis of ATP or ADP and is in dynamic contact with neurons, astrocytes, and vascular segments. SMCs are located in precapillary arteries, and glutamate released by neurons can directly bind to the abundant functional GluN1 receptors on their surface to mediate vasodilation. Pericytes are mainly located in capillary segments covering ECs and embedded in the basement membrane, spreading discontinuously along microvessels and maintaining the barrier properties of the BBB.

**Figure 2 cells-14-00016-f002:**
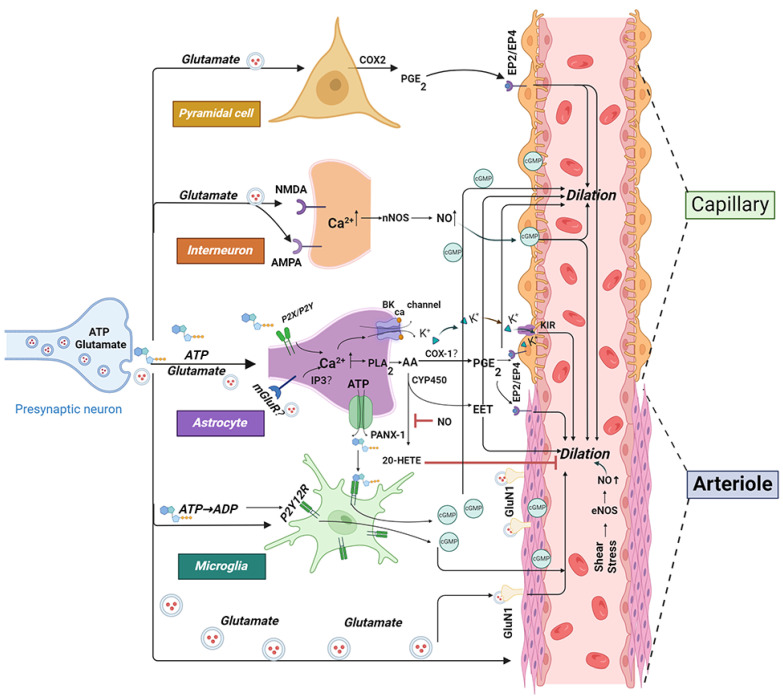
Hypothetical signaling pathways and mechanisms of neurovascular coupling. AA, arachidonic acid; AMPA:α-amino-3-hydroxy-5-methyl-4-isoxazolepropionic acid; ADP, adenosine diphosphate; ATP, adenosine triphosphate; cGMP, cyclic guanosine monophosphate; EET, epoxyeicosatrienoic acid; COX, cyclooxygenasenase; CYP, cytochrome P450; 20-HETE, 20-Hydroxyeicosatetraenoic acid; EP, prostaglandin E2 receptor; NMDAR, N-methyl-D-aspartate receptor; NO, nitric oxide; PG, prostaglandin; PLA2, phospholipase A2; purines from pannexin 1:PANX1. mGluR5, metabotropic glutamate receptor 5; NOS, nitric oxide synthase. “?” It represents the controversial point in the current research. Red line represents inhibition. The black arrow line represents promotion or elevation.

**Figure 3 cells-14-00016-f003:**
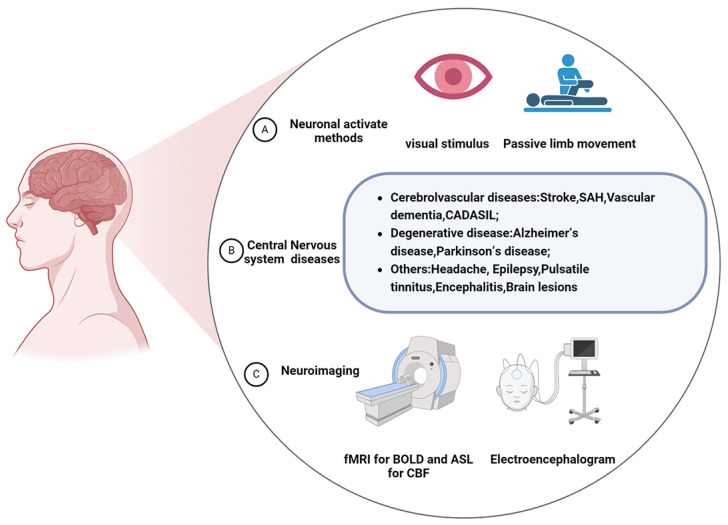
Clinical assessment of NVC in central nervous system diseases.

**Table 1 cells-14-00016-t001:** Cellular mechanism of neurovascular coupling.

Mechanism Association	Specific Pathway	Animal	Agent or Method	Stimulation	NVC Accessed with	Study
Neuron	acetylcholine	SD rats	cytochrome P450 inhibition, COX-2 inhibition, EETs antagonist	BF stimulation	In vivo LDF and ECoG	Lecrux C et al., 2017 [17]
Neuron	nNOS	C57BL/6J	Virus injections	Limb stimulation	In vivo TPLSM and microelectrode ex vivo section	Echagarruga CT et al., 2020 [22]
Neuron	GABA	SD rats	Cholinergic deafferentation and pharmacological manipulations	BF electrical stimulation and pharmacological stimulation	In vivo LDF and microelectrode	Kocharyan A et al., 2008 [19]
Neuron	COX-2	Wistar or SD rats	U46619, NMDA, SC 560, NS 398, AH 6809 and L 161–982	Whiskers stimulation and optogenetic stimulation	In vivo LSI or LDF and ex vivo patch clamp	Lacroix A et al., 2015 [26]
Neuron	NPY-Y1 pathway	C57Bl6/ICR	AP5, CNQX, TTX and BIBP	Forepaw electrical stimulation and optogenetic stimulation	In vivo TPLSM and microelectrode	Uhlirova H et al., 2016 [27]
Neuron	Ca^2+^	SD rats	Virus injections	Forepaw stimulation	In vivo multimodality optical imaging platform and ex vivo section	Chen W et al., 2020 [75]
Neuron	Ca^2+^	C57BL	Viral injection	Awake and anesthetized states	In vivo Multi-modality optical imaging	Park K et al., 2018 [76]
Neuron	nNOS, STT	C57Bl/6J	Transgene	Whisker stimulation and optogenetic stimulation	In vivo 2D-OIS, fMRI and microelectrode	Lee L et al., 2020 [77]
Interneurons	Ca^2+^ response	C57BL/6J	Viral injection	Optogenetic stimulation	In vivo TPLSM and microelectrode	Krogsgaard A et al., 2023 [78]
Astrocyte	Ca^2+^ response	Wistar rats	No	Electrical forepaw stimulation	In vivo LDF and ex vivo section	Zonta M et al., 2003 [29]
Astrocytes	NMDAR	Grin1fl/fl mice	Chemogenetically induced	Pharmacology stimulation	In vivo TPLSM and ex vivo section	Lu L et al., 2019 [79]
Astrocyte	Ca^2+^ response	C57BL/6J	Transgene	Afferent fibers electrical stimulation	In vivo IOS and TPLSM and ex vivo section	Institoris A et al., 2022 [34]
Astrocyte	Ca^2+^ response	C57BL/6J	Transgene and Viral injection	Electrical forelimb stimulation	In vivo TPLSM and LDF	Gu X et al., 2018 [39]
Astrocytes	Ca^2+^ response	C57BL6 mice and Wistar rats	Transgene	Pharmacology stimulation	Ex vivo section	Otsu Y et al., 2015 [80]
Astrocytes	Ca^2+^ responses	Crl:NMRI	Chemogenetically induced	Trigeminal nerve electrical stimulation	In vivo fEPSP and IOS or LSI	Lind BL et al., 2018 [81]
Microglia	P2Y12R	C57BL/6J	Chemogenetically induced	Whisker stimulation	In vivo TPLSM, LSI, fUS, microdrive, SPECT and PET imaging	Császár E et al., 2022 [41]
Microglia	PANX1-P2RY12	P2RY12−/−:CX3CR1GFP/+mice	Transgene	No	In vivo TPLSM	Bisht K et al., 2021 [71]
Microglia	P2Y12R	C57BL/6	Transgene	No	In vivo TPLSM and STORM super-resolution imaging; ex vivo section	Cserép C et al., 2020 [40]
Glio-endothelial	P2Y1R, eNOS	C57BL/6J	eNOS deficient	Whisker stimulation	In vivo LDF and ex vivo blood vessel	Toth P et al., 2015 [43]
ECs	Kir2.1 channel	Col4a1 mutant mouse	Model of cSVD	Whisker stimulation	In vivo LDF and ex vivo section electrophysiology	Thakore P et al., 2023 [82]
ECs	NMDAR	C57Bl/6J	grin1fl/fl · Cre+/− (Cre) mice with eNMDAR loss of function	Whisker stimulation	In vivo TPLSM and LDF	Hogan-Cann AD, et al., 2019 [45]
ECs	NMDAR	C57BL/6	tPA wt or tPA−/−	Whisker stimulation	In vivo LSI and fUS	Anfray A et al., 2020 [51]
ECs	GqPCR-PIP2-Kir2.1	C57BL / 6J	/	Chemical stimulation	In vivo multiphoton microscope; ex vivo capillary-parenchymal arteriole for patch-clamped	Harraz OF et al., 2018 [57]
ECs	KIR2.1 channel	C57/BL6J	KIR2.1 gene KO	Whisker stimulation	In vivo LDF, microelectrode, and multiphoton microscope; ex vivo section for patch-clamped	Longden TA, et al., 2017 [37]
ECs	TRPA1 channels	C57BL/6J	Trpa1-ecKO	Whisker stimulation	In vivo LSI and TPLSM	Thakore P, et al., 2021 [50]
ECs	Caveolae	C57BL/6J	Transgene	Whisker stimulation	In vivo TPLSM and ex vivo section	Chow BW et al., 2020 [58]
SMC	K-ATP channel	mice, zebrafish, and cell culture models	Kcnj8 KO and transgene	Whisker stimulation	In vivo TPLSM and ex vivo pericytes	Ando K et al., 2022 [83]
Pericytes	PDGFR-β	Pericyte-CreER mice	Chemogenetically induced	Hind-limb stimulation	In vivo LDF, TPLSM, IOS and VSD Imaging	Kisler K, etal., 2022 [69]
Pericytes	/	PDGFRβ-ChR2-YFP mice	Optical ablation of pericytes	Optogenetic stimulation	In vivo TPLSM	Hartmann DA et al., 2021 [72]
Pericytes	NO-20-HETE	SD or Wistar rats and NG2-DsRed C57BL/6J mice	No	Whisker stimulation	In vivo IOS and ex vivo pericytes for patch-clamped	Hall CN et al., 2014 [38]
Pericytes	Kir2.1 channel	NG2-DsRed-BAC C57BL/6J mice	Transgene	Pharmacology stimulation	In vivo LDF, microelectrode, and multiphoton microscope; ex vivo section for patch-clamped	Grubb S et al., 2020 [54]
Pericytes	/	129S1/SvlmJ and C57BL/6J mice	Transgene	Hind-limb stimulation	In vivo LDF, IOS, TPLSM and microelectrode; ex vivo section	Bazargani N et al., 2016 [2]
Pericytes	/	Pericyte-CreER mice	Chemogenetically induced	No	In vivo LDF, MRI and TPLSM; capillary isolation and primary cell cultures	Nikolakopoulou AM et al., 2019 [70]

“/” indicates specific pathway studies or agent interventions not covered by the study. arterial spin labelling: ASL; ATP sensitive potassium channel: KATP; arteriolar endothelial cells: aECs; basal forebrain: BF; common carotid artery occlusion: CCAO; cyclization recombinase: Cre; cyclo-somatostatin: C-SOM; central nervous system: CNS; endothelial NO synthase: enhanced green fluorescent protein: eGFP; electrocorticogram: ECoG; electrogenic sodium-bicarbonate cotransporter 1:NBCe1; epoxy-eicosatrienoic acids: EETs; eNOS; endothelial cell: EC; field excitatory postsynaptic potential: fEPSP; insulin receptors: IRs; intrinsic optical signals: IOS; local field potential: LFP; laser Doppler flowmetry: LDF; multi-unit activity: MUA; magnetic resonance imaging: MRI; neurovascular coupling: NVC; Neuropeptide Y: NPY; optical intrinsic signal: OIS; parenchymal arteriole: PA; reactive oxygen species: ROS; signal-to-noise ratio: SNR; smooth muscle: SMC; soluble epoxy hydrolase: sEH; somatostatin: SST; soluble epoxy hydrolase: sEH.

**Table 2 cells-14-00016-t002:** Clinical research experimental methodologies and results.

Diseases	Participants (N)	Stimulation	NVC Accessed with	Study
Stroke	17 IV-tPA within 6 h after stroke symptoms begin or with embolectomy	No	TCD for CBF and EEG for neuronal activity	Wu D et al., 2017 [84]
HCs	12 healthy subjects	Visual task to activate the visual cortex	Measuring posterior and MCA blood velocity through in software	O’Gallagher K et al., 2022 [85]
CADASIL	27 CADASIL patients and 20 HCs	Visual task to activate the visual cortex	TCD for CBFV, CA and VR; and capnograph for EtCO2	Jokumsen-Cabral A et al., 2019 [86]
Stroke	15 mild, 27 moderate and 13 severe stroke patients, and 32 control subjects	1 min passive flexion and extension of the elbow	TCD for CBFV and capnograph for EtCO2	Sutherland BA et al., 2017 [87]
CADASIL	19 CADASIL patients and 19 HCs	Visually cued motor tasks	ASL-fMRI for BOLD and EEG for neuronal activity	Huneau C etal., 2018 [88]
HCs	31 young 11 female and 32 older adults	Trail making task	TCD for CBF and capnograph for EtCO2	Toth L et al., 2022 [89]
HCs	10 healthy young adults	Finger tapping task	TCD for CBF and fNIRS for neuronal activity	Csipo T et al., 2021 [90]
SIVD	24 normal controls and 54 patients with SIVD	No	fMRI for ReHo and CBF	Liu X et al., 2021 [91]
HCs	18 healthy young adults	Passive motor stimulation	TCD for CBF, capnograph for EtCO2 and ECG for neuronal activity	Maggio P et al., 2014 [92]
HCs	20 healthy adult volunteers	During an afternoon nap	fMRI for BOLD and EEG for neuronal activity	Betta M et al., 2021 [93]
HCs	20 healthy men	Hypoxic stimulation	TCD for CBF	Lefferts WK et al., 2016 [94]
PI	49 patients with PI and 30 matched normal subjects	No	ASL for CBF and fMRI for FCS	Wang P et al., 2023 [5]
AD	15 patients with ADD; 24 patients with MCI; 15 cognitively HC	Ficker stimulation	Dynamic Vessel Analyzer for vessel	Shang S et al., 2022 [95]
CM	16 CM patients and 11NCs	No	fMRI for BOLD and ASL	Hu B et al., 2019 [96]
IGE	26 children with IGE and 35 HCs	No	fMRI and ASL for DC and CBF	Hu J et al., 2023 [97]
HCs	64 volunteer subjects	Auditory stimulation	fNIRS for hemodynamic and EEG for neuronal activity	Muñoz V et al., 2023 [98]
Encephalitis	23 anti-NMDAR encephalitis patients and 30 HCs	No	fMRI for CBF-ReHo and fALFF-CBF	Guo Y et al., 2022 [99]
MOH	40 patients with MOH and 32 HCs	No	fMRI for ReHo-CBF, fALFF-CBF and DC-CBF	Li X et al., 2023 [100]
HCs	13 MS patients and in 10 HCs	Visual Stimulation	MEG for neural activity and fMRI for CBF and BOLD	Stickland R et al., 2019 [101]
PT	24 right PT patients and 25 HCs	No	fMRI for ReHo and ASL for CBF	Li X et al., 2022 [102]
HCs	4832 Chinese Han subjects	No	fMRI for BOLD and ASL for CBF	Xue H et al., 2023 [103]
Stroke	13 patients with LIAS, 21 with SVD and 17 HCs	Visual Stimulation	TCD for CBF	Lin WH et al., 2011 [104]
Stroke	8 recovered patients from stroke and 8 HCs	Motor task and visual task	fMRI for BOLD	Krainik A et al., 2005 [105]
aSAH	70 aSAH patients and 42 HCs	three cycles of flicker-light excitation	Non-invasive retinal vessel analysis	Albanna W et al., 2021 [106]

Acute ischemic stroke: AIS; Atrial fibrillation: AF; aSAH: aneurysmal subarachnoid hemorrhage; cerebral blood flow: CBF; cerebral blood flow velocity: CBFV; cerebral autoregulation: CA; intravenous injection of tissue plasminogen activator: IV-tPA; degree centrality: DC; degree centrality: DC; End-tidal CO2: EtCO2; Electroencephalography: EEG; fractional amplitude of low-frequency fluctuation: fALFF; functional near-infrared spectroscopy: fNIRS; functional connectivity strength: FCS; healthy controls: HCs; idiopathic generalized epilepsy: IGE; mild cognitive impairment due to AD: MCI; medication-overuse headache: MOH; middle cerebral arteries: MCA; Multiple Sclerosis: MS; mild-to-moderate dementia due to AD: ADD; middle cerebral arteries occlusion: MCAO; neurovascular unit: NVU; regional homogeneity: ReHo; pontine infarction: PI; Parkinsons disease: PD; pulsatile tinnitus: PT; sleep deprivation: SD; subcortical ischemic vascular disease: SIVD; vasoreactivity: VR; transcranial Doppler: TCD.

## Data Availability

No new data were created or analyzed in this study. Data sharing is not applicable to this article.

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
