# Peer review of "From Mechanisms to Medicine: Neurovascular Coupling in the Diagnosis and Treatment of Cerebrovascular Disorders: A Narrative Review"

_cells, 2024, doi:10.3390/cells14010016_

Round 1

Reviewer 1 Report

Comments and Suggestions for Authors

interesting manuscript as review but there are some supplemental files that could be in the print.

Author Response

Comment 1: interesting manuscript as review but there are some supplemental files that could be in the print.

Response: Thanks for the kind suggestion. We have put the supplementary materials in the print according to your suggestion. (page 3, lines 94-104).

Reviewer 2 Report

Comments and Suggestions for Authors

This is an interesting review that discusses the role of NVC in cerebrovascular disorders. 

The authors should edit the manuscript for numerous cases when the literature references should be mentioned. 

For example, the following sentences should be followed by a reference: "When neurons in a specific area are active, they can trigger vasodilation directly by stimulating vascular smooth muscle cells and pericytes or indirectly through astrocytes, thus rapidly increasing local cerebral blood flow."  "Neurons, either directly or through astrocytes, signal SMCs and pericytes to dilate nearby blood vessels. This response increases blood flow precisely where it is most needed to support the metabolic demands of active brain regions." These sentences are just examples of sentences where the literature references are needed. There are many other cases in the text of the manuscript where the authors should provide references. 

Some sentences need to be edited for logical or meaningful understanding. For example, the authors should edit the sentence "A deeper understanding of the mechanisms of cellular interactions in NVC allows for treating NVC as the smallest unit, enabling neurological diseases to be addressed at a global level and providing new ideas for improving impaired neurological symptoms." for better understanding by readers. 

In some cases the literature references numbers should be unified in style mentioning. For example in next two sentences "This advanced coordination is critical for the effectiveness of modern brain imaging techniques, which rely on vascular signals as proxies for neuronal activity." and "Mechanical shear stress can induce the release of NO from endothelial NOS (eNOS) in ECs, which acts on guanylate cyclase, thereby leading to the formation of cyclic guanosine monophosphate (cGMP), smooth muscle relaxation, and arteriolar dilation [24, 25].24,25" the references numbered in upper style.

The authors should describe in more detail the explanation for Tables 1 and 2 content and explain why these particular publications were chosen for Tables.

In the following sentence  "In concert with the well-known microvascular proximity of interneurons and their processes, these observations suggest distinct mechanisms through which neural activity increases cholinergic basal forebrain (CBF)" the authors provided CBF abbreviation which is better to use for "Cerebral Blood Flow" words that mentioned more often through the text of the manuscript.

The typo errors should be eliminated (for example "Ischemia Stroke" should be edited as "Ischemic Stroke"

Author Response

Comment 1: The authors should edit the manuscript for numerous cases when the literature references should be mentioned. For example, the following sentences should be followed by a reference: "When neurons in a specific area are active, they can trigger vasodilation directly by stimulating vascular smooth muscle cells and pericytes or indirectly through astrocytes, thus rapidly increasing local cerebral blood flow."  "Neurons, either directly or through astrocytes, signal SMCs and pericytes to dilate nearby blood vessels. This response increases blood flow precisely where it is most needed to support the metabolic demands of active brain regions." These sentences are just examples of sentences where the literature references are needed. There are many other cases in the text of the manuscript where the authors should provide references.

Response 1: Thanks for the kind suggestion. We have edited the manuscript for numerous cases when the literature references should be mentioned based on your suggestion.They are the references in red font in the manuscript. Such as page 1, line 33; page 2, line 75; page 4, lines 148-149; page 8, lines 176-177; page 11, line 336; page 12, lines 392, 394; page 12, lines 416, 426; page 17, lines 546, 572, 574; page 18, line 601; page 18, line 609, 618, 619, 622, 641, 642; page 19, line 650, 666.

Comment 2: Some sentences need to be edited for logical or meaningful understanding. For example, the authors should edit the sentence "A deeper understanding of the mechanisms of cellular interactions in NVC allows for treating NVC as the smallest unit, enabling neurological diseases to be addressed at a global level and providing new ideas for improving impaired neurological symptoms." for better understanding by readers.

Response 2: Thanks for the kind suggestion. We've rewritten the phrase as follows: “This approach allows for a holistic treatment of neurological diseases and offers innovative strategies to ameliorate compromised neurological symptoms ”. This will give the reader a better understanding of the meaning.

Comment 3: In some cases the literature references numbers should be unified in style mentioning. For example in next two sentences "This advanced coordination is critical for the effectiveness of modern brain imaging techniques, which rely on vascular signals as proxies for neuronal activity." and "Mechanical shear stress can induce the release of NO from endothelial NOS (eNOS) in ECs, which acts on guanylate cyclase, thereby leading to the formation of cyclic guanosine monophosphate (cGMP), smooth muscle relaxation, and arteriolar dilation [24, 25].24,25" the references numbered in upper style.

Response 3: Thanks for the kind suggestion. We have changed the citation formatting of the references to uppercase in response to your comments. (page 2, line 82; page 4, line 142; page 9, line 269).

Comment 4: The authors should describe in more detail the explanation for Tables 1 and 2 content and explain why these particular publications were chosen for Tables.

Response 4: Thanks for the kind suggestion. We described in more detail the explanation for Table 1 as follows: “We next elaborate on the mechanism of action of each cell type within the NVU in NVC, which display representative studies of cellular mechanisms of NVC, mainly showing the cells involved in studying NVC, animal models, stimulation modalities, imaging techniques in Table 1.”(page 4, line150-153). We also described in more detail the explanation for Table 2 as follows: “We display in Table 2 the ways in which NVC is induced in various clinical diseases, assessment tools, etc.”(page 14, lines 521-522).

Comment 5: In the following sentence  "In concert with the well-known microvascular proximity of interneurons and their processes, these observations suggest distinct mechanisms through which neural activity increases cholinergic basal forebrain (CBF)" the authors provided CBF abbreviation which is better to use for "Cerebral Blood Flow" words that mentioned more often through the text of the manuscript.

Response 5: Thanks for the kind suggestion. Based on your comments, we have identified “CBF” in the article as an abbreviated form of cerebral blood flow. (page 1, line 33).

Comment 6: The typo errors should be eliminated (for example "Ischemia Stroke" should be edited as "Ischemic Stroke"

Response 6: Thanks for the kind suggestion. We have corrected the misspellings based on your suggestions. (page 16, line 514).

Reviewer 3 Report

Comments and Suggestions for Authors

The review discusses the cellular mechanisms underpinning NVC, its impairment in various cerebrovascular disorders, and the potential diagnostic and therapeutic strategies targeting NVC dysfunction. I have several major and minor concerns with this manuscript.

1. Based on the iThenticate report indicating a 46% overlap, significant portions of the manuscript may have been previously published or are heavily reliant on existing literature without sufficient novel interpretation or synthesis. 

2. The manuscript could benefit from a more critical analysis of recent advancements in imaging technologies and their impact on understanding NVC. Additionally, discussing emerging therapeutic strategies with a focus on innovative approaches could add value.

3. The review discusses various cellular components involved in NVC but does not sufficiently address the complex interactions between these components in pathological states. A more integrated analysis could provide a better understanding of how these interactions contribute to disease. 

4. The manuscript covers a broad range of topics related to neurovascular coupling, from basic cellular mechanisms to clinical implications. While this breadth is valuable, it can dilute the depth of discussion on specific issues, particularly the clinical translation of basic research findings.

5. The manuscript could be strengthened by proposing specific future research directions, especially in areas where current knowledge is limited or controversial.

6. The review sometimes lacks a clear structure, making it challenging to follow the progression from basic mechanisms to clinical applications. 

7. there are several areas  where improvements in grammar and comprehension could enhance clarity and readability

8. Throughout the manuscript, ensure that technical terms are defined when first used, especially if they are not common knowledge even among specialists. This will help improve comprehension for a broader scientific audience.

9. Ensure consistent use of terminology throughout the manuscript. For instance, if abbreviations like "CBF" (cerebral blood flow) are used, they should be defined once and used consistently.

Comments on the Quality of English Language

The manuscript needs significant editing for language and comprehension.

Author Response

The review discusses the cellular mechanisms underpinning neurovascular coupling, its impairment in various cerebrovascular disorders, and the potential diagnostic and therapeutic strategies targeting neurovascular coupling dysfunction. I have several major and minor concerns with this manuscript.

Comment 1: Based on the iThenticate report indicating a 46% overlap, significant portions of the manuscript may have been previously published or are heavily reliant on existing literature without sufficient novel interpretation or synthesis.

Response 1: Thank you very much for the valuable comments. We have made changes based on the suggestions and checked again and the iThenticate report shows about 25%.

Comment 2: The manuscript could benefit from a more critical analysis of recent advancements in imaging technologies and their impact on understanding NVC. Additionally, discussing emerging therapeutic strategies with a focus on innovative approaches could add value.

Response 2: Thanks for the kind suggestion. We provided a more critical analysis of recent advances in imaging technology and their implications for understanding NVC in Part 4. The main points are as follows: “Current advances in imaging techniques for nerve cells and CBF allow us to observe, manipulate, and understand the cellular activity of members of the NVU in greater depth. We can observe how NVC affects overall brain function and disease progression by studying NVC in disease through these imaging techniques ”.(page 13, lines 472-475). Functional magnetic resonance imaging (fMRI), based on BOLD uses the difference between oxygenated and deoxygenated hemoglobin in response to a magnetic field to form different signals that indirectly reflect the functional activity of neurons [99]. It can objectively reflect changes in the activity of various functional areas of the brain in real time. Arterial spin labeling (ASL) refers to arterial spin labeling of blood flowing into the brain through a radiofrequency pulse [100]. Based on the signal difference in blood, information on cerebral blood perfusion is obtained and can be calculated from the quantitative CBF data. High temporal resolution electroencephalography (EEG) and high spatial resolution functional near-infrared imaging (fNIRS) complement each other and facilitate a more comprehensive study of NVC mechanisms in the brain during working memory [101]. These techniques allow NVC to be evaluated in both time and space and it's already being used in the clinic (Figure 3). “Despite advances in imaging techniques, some limitations and drawbacks remain. For example, oxygenation-based methods, such as fMRI and NIRS, only indirectly measure functional hyperemia, which may limit a comprehensive understanding of NVC mechanisms. ASL, while it can measure CBF changes, has lower temporal resolution than using BOLD contrast. In summary, imaging technologies provide powerful tools for NVC research, but technical challenges and limitations still need to be overcome to achieve a more comprehensive understanding of NVC. Future research needs to further optimize these techniques to improve imaging depth, resolution, and temporal resolution while reducing cost and technical thresholds. We display the stimulation methods and assessment tools related to NVC in clinical diseases in Table 2. We will now move from the cellular level to a more macroscopic view of NVC research in specific diseases. This will help us understand the specific manifestations of NVC in pathological states and be crucial for developing therapeutic strategies for specific diseases. ”(pages 13-14, lines 487-499).

Additionally, based on your suggestions, we focused our discussion on interventions for NVC around innovative approaches in part 5. The main points are as follows: “A study treated 24-month-old C57BL/6 mice with NMN, a NAD intermediate, for two weeks, finding that NMN supplementation improved NVC responses by enhancing endothelial NO-mediated vasodilation, leading to better spatial working memory and gait coordination [151]. This suggests that reduced NAD availability contributes to age-related cerebromicrovascular dysfunction and cognitive decline, making NAD intermediate potential interventions for vascular cognitive impairment. Supporting this, PJ-34 and SS-31 were shown to improve cerebromicrovascular endothelial function and cognitive performance in aged mice [152, 153]. mTOR, which drives cerebrovascular dysfunction in AD models by reducing eNOS activity, can be targeted with rapamycin to restore NVC function [154, 155]. Oxidative stress-induced NVC dysfunction, implicated in VaD, was mitigated by LG, which may protect NO bioavailability and maintain NVC sensitivity [156]. Additionally, tPA in circulation was found to influence NVC by potentiating CBF increase through a tPA-dependent activation of NMDAR on endothelial cells [67].

According to the latest research findings about the mechanism of NVC, some potential future intervention targets and innovative intervention approaches have also emerged. Research by Jiemin Jia's team at Westlake University has revealed the mechanism of direct dialog between neurons and cerebral blood vessels, a finding that provides new therapeutic ideas for NVC dysfunction [4]. They found that there are “holes” in the wrapping rate of astrocyte end-feet to the peripheral arteries, and the presynaptic sub-synapses of perivascular neuron axons will cross these holes and form “synapse-like” structures with vascular smooth muscle cells, which provide a new bridge for the dialog between neurons and blood vessels [4]. This structure provides a new bridge for the dialog between neurons and blood vessels. Using optogenetic activation techniques, the team demonstrated that this structure can regulate vasoconstriction and diastole in the brain, thereby affecting NVC [4].

The NsMJ structure allows glutamatergic neurons to act directly on arterial vascular smooth muscle cells, leading to arteriolar diastole and inducing functional cerebral congestion [4]. This finding provides new insights into understanding the rapid and precise regulation of CBF and offers new ideas for the effective treatment of ischemic hypoperfusion injury. The team also proposed a comprehensive and systematic vascular dynamics analysis algorithm, which not only describes the basic characteristics of cerebral arterial vasomotion in mice under physiological conditions but also records and analyzes in real-time the damage and recovery of cerebral arterial vasomotion and CBF encounters in ischemic stroke conditions in mice [172]. Reactivation of spontaneous vasomotion in cerebral arteries by transgenic (ME-Linker) mice achieves optimal reperfusion while ameliorating neurological damage and contributing to cerebrovascular recovery after ischemic stroke [172]. These research advances suggest that new therapeutic approaches can be developed through an in-depth understanding of the molecular and cellular mechanisms of NVC, thereby improving the therapeutic outcomes of neurological diseases such as stroke. These results not only deepen our understanding of the role of NVC in the regulation of CBF but also provide a scientific basis and a new direction for future clinical treatment.

In addition, research by Yan Dou's group at Tianjin Medical University General Hospital has proposed a cross-scale targeted nanotherapy based on the natural self-assembling SIRT1 activator Ren [173]. This approach achieves precise and efficient treatment of Alzheimer's disease by specifically targeting damaged cerebral microvascular endothelial cells, utilizing a short peptide of the receptor for advanced glycosylation end products (RAP) coupled with an amphiphilic block copolymer, PLGA-PEG, and further self-assembled with ellagic acid [173]. Nitric oxide releases and improves the integrity of the blood-brain barrier, while promoting neuronal mitochondrial biogenesis and the conversion of glucose metabolism to oxidative phosphorylation, thus remodeling the neurovascular and neurometabolic coupling [173]. Their study reveals a direct dialog mechanism between neurons and cerebral blood vessels, a finding that is not only theoretically important but also shows potential clinical applications. The potential value of these innovative approaches lies in the fact that they not only provide a new understanding of dysfunctional NVC but also provide a scientific basis for the development of new therapeutic approaches. As research progresses, these innovative interventions are expected to play an important role in future clinical practice.”(pages 20-22, lines 731-829).

Comment 3: The review discusses various cellular components involved in NVC but does not sufficiently address the complex interactions between these components in pathological states. A more integrated analysis could provide a better understanding of how these interactions contribute to disease.

Response 3: Thanks for the kind suggestion. It is not yet clear how each cell interacts with each other to cause disease in a given disease state. However, in Part 3, we describe in detail the role of each cell in the NVC process and how they coordinate with each other to ensure normal NVC. We have made some comprehensive analyses based on the cellular interactions of NVC and the pathological changes of cells in disease states as follows: “Currently, NVC-related research focuses on two major modules: one is to conduct indepth research on the mechanism of a cell's role in the NVC; the other is to explore whether the phenomenon of neurovascular uncoupling occurs after the onset of disease, and the potential mechanism and whether this phenomenon affects neurological function. The cellular components of the NVU may lead to neurovascular uncoupling through complex interactions. Local CBF is directly affected when neuronal activity is reduced or abnormal. Neurons affect the peripheral vascu-lature by releasing metabolites and neurotransmitters such as glutamate, whereas astrocytes and microglia are involved in regulating vascular responses by releasing vasoactive substances such as prostaglandins and NO [38, 47]. In pathological states, changes in the reactivity of these cells may lead to vasodilatory dysfunction, which in turn affects CBF. The altered activation state of astrocytes may lead to an imbalance in their release of vasoactive substances, whereas overactivation of microglia may exacerbate inflammatory responses and vascular injury [57]. Dysfunction of VSMCs and ECs may lead to reduced vascular reactivity and BBB disruption, further affecting NVC [20, 27]. Abnormal contraction of pericytes at the capillary level may lead to increased vascular permeability, exacerbating brain edema and inflammation [50]. Together, these factors lead to altered vascular reactivity, disruption of the BBB, hemodynamic disturbances, and increased neuroinflammation, ultimately leading to uncoupling of NVC and affecting brain tissue repair and functional recovery. However, these are just some basic theories, which do not have much guiding value for clinical translation. Future research should focus on exploring how much damage to neurons or other cellular components causes neurovascular uncoupling, and how much neurovascular uncou-pling causes neurological dysfunction. This will guide us in the search for targets for intervention and the application of therapeutic measures.”(pages 13, lines 458-481).

Comment 4: The manuscript covers a broad range of topics related to neurovascular coupling, from basic cellular mechanisms to clinical implications. While this breadth is valuable, it can dilute the depth of discussion on specific issues, particularly the clinical translation of basic research findings.

Response 4: Thanks for the kind comments. We take into account that a large number of current studies still focus on the role of a particular cell of the neurovessel, which is rather one-sided. In addition, other studies, even though they focus on the integrity of the entire neurovascular unit, do not test the collaborative function of the neurovascular unit. Our aim in writing this manuscript is to review and re-emphasize the importance of NVC in CNS diseases, especially cerebrovascular disease. To bring the attention of more researchers and scholars to the importance of studying the NVC as a whole in diseases. Thus the content of our manuscript is broad in its coverage. Based on the comments 2, 3, and 5 suggestions. We have enhanced our indepth discussion of neurovascular coupled imaging techniques, cellular interactions in neurovascular coupled disease states, innovative therapeutic modalities, as well as current research controversies, and future research directions.

Comment 5: The manuscript could be strengthened by proposing specific future research directions, especially in areas where current knowledge is limited or controversial.

Response 5: Thanks for the kind comments. We have enhanced the description of future research directions in NVC and where current NVC research is unclear and controversial as follows: “There are several controversies and ambiguities in current NVC research, and these are mainly centered on the regulatory mechanisms, influencing factors, and relationship with disease in NVC. First, in terms of the regulatory mechanisms of NVC, although neuronal activity is known to be able to influence CBF through multiple pathways, the specific molecular and cellular mechanisms remain incompletely understood. For example, the roles of astrocytes and microglia in NVC are controversial, and they may regulate CBF by releasing vasoactive substances or participating in inflammatory responses, but the specific mechanisms of action and the extent of the effects are yet to be further investigated. Secondly, the interaction between NVC and the BBB is also a hot research topic, and the disruption of the BBB may affect NVC, but the specific roles and regulatory mechanisms of the BBB in NVC remain unclear. In addition, the changes in NVC in different brain regions and different disease states are controversial. For example, in diseases such as Alzheimer's disease and depression, it is inconclusive whether the changes in NVC are the cause, the result, or both. At the technical level, while in vivo imaging techniques can provide information on the dynamics of NVC, their spatiotemporal resolution limits the capture of rapid neurovascular events, and while in vitro experiments can provide detailed information at the molecular level, how to correlate this information with in vivo status remains a challenge. Finally, the individual variability of NVC is also an area of ambiguity; differences in NVC between individuals may influence disease onset and progression, but the understanding and control of these differences are currently limited.” (page 22, lines 846-866) In addition, each component of the NVU has a vital role in the transmission of information. What are the contributions of neurons, glial cells, and vascular recoupling? Which of the perforating arteries, precapillary arteries, or capillaries, is the leading segment that regulates cerebral microvascular blood flow? These are not established in current research. The cellular communication involved and the underlying molecular mechanisms are complex. Large-scale omics, imaging markers, and novel molecular biomarkers are immediately required for further mechanistic exploration.

In addition to the examples discussed here, there is emerging evidence for the functional regulation of NVC throughout life activities. For example, sleep and circadian mechanisms appear to regulate barrier permeability, at least in invertebrate models. It would be interesting to determine if and how these findings apply to mammals. However, it is not clear whether neural activity causes vasomotor activity and vice versa, or whether there is only a correlation. Additionally, vascular permeability and blood flow can feed back to each other, particularly in aging individuals. Sensitive real-time monitoring of vasoconstriction and relaxation, as well as real-time stimulation and imaging of neural activity, may help elucidate this relationship. In addition, the growing toolkit of optogenetics, chemogenetics, and biosensors provides a means for controlling and detecting the activities of living animals. These methodologies have the potential to address several critical questions in the field of NVC. The primary challenge moving forward is to develop therapeutic strategies to address neurovascular dysfunction by targeting impaired NVC, thereby preventing disease progression. Changes in NVC may vary over time, especially in neurodegenerative diseases. It is also a challenge to develop techniques that allow stable monitoring of NVC over time. (page 23, lines 886-888).

Comment 6: The review sometimes lacks a clear structure, making it challenging to follow the progression from basic mechanisms to clinical applications.

Response 6: Thanks for the kind suggestion. We have added some descriptions to the article based on your suggestions to make the transition from the cellular mechanisms of NVC to the disease more fluid as follows: “Having detailed the cellular mechanisms of NVC, we have gained a deep understanding of the molecular and cellular level changes involved in this complex physiological process. This basic science knowledge has shed light on the central role of NVC in maintaining brain function and provided us with a starting point for exploring its changes in disease states.” (page 13-14, lines 484-488).

Comment 7: there are several areas where improvements in grammar and comprehension could enhance clarity and readability

Response 7: Thanks for the kind suggestion. We have asked experts to revise the grammar of the language to some extent, hoping to help the overall understanding of the article.

Comment 8: Throughout the manuscript, ensure that technical terms are defined when first used, especially if they are not common knowledge even among specialists. This will help improve comprehension for a broader scientific audience.

Response 8: Thanks for the kind suggestion. We've rechecked the entire text and defined terms that appear for the first time for ease of understanding. For example, let's replace “Virchow-Robin” with “perivascular spaces”, which have the same meaning and are easier to understand. (page 2, lines 65 and 66). In addition, We provide further explanatory notes on the terminology of local field potentials as follows: “LFP refers to weak electrical signals generated by the synchronized activity of a large number of neurons in the cerebral cortex or neural tissue. These electrical signals are generated by the superposition of postsynaptic potentials (mainly excitatory and inhibitory postsynaptic potentials) of neurons, and they reflect the activity state of the neuronal population.”(page 8, lines 199-203).

We also provide further explanatory notes on the terminology of diffuse depolarization (SD) as follows: “SD is caused by local ischemia of brain tissue, where the ionic balance of neurons and glial cells is disrupted, leading to extensive depolarization of cell membrane potentials.”(page 17, lines 546-548).

Comment 9: Ensure consistent use of terminology throughout the manuscript. For instance, if abbreviations like "CBF" (cerebral blood flow) are used, they should be defined once and used consistently.

Response 9: Thanks for the kind suggestion. We have identified “CBF” in the article as an abbreviated form of cerebral blood flow based on your comments. (page 1, line 33).

Round 2

Reviewer 2 Report

Comments and Suggestions for Authors

The authors significantly improved the manuscript, but it is still needed for additional edits. 

For example: "We next elaborate on the mechanism of action of each cell type within the NVU in NVC, which displays representative studies of cellular mechanisms of NVC, mainly showing the cells involved in studying NVC, animal models, stimulation modalities, and imaging techniques in Table 1" need to be edited as "We next elaborate on the mechanism of action of each cell type within the NVU in NVC, which displays selected representative studies of cellular mechanisms of NVC, mainly showing the cells involved in studying NVC, animal models, stimulation modalities, and imaging techniques in Table 1"

The authors should explain what means "/" symbol in Table 1?

"Ach" should be edited as "ACh".

 "aECs" abbreviation should be used in the first mention.

"For example, sleep and circadian mechanisms appear to regulate barrier permeability, at least in invertebrate models."-"barrier" should be explained

Author Response

Comment 1:The authors significantly improved the manuscript, but it is still needed for additional edits. For example: "We next elaborate on the mechanism of action of each cell type within the NVU in NVC, which displays representative studies of cellular mechanisms of NVC, mainly showing the cells involved in studying NVC, animal models, stimulation modalities, and imaging techniques in Table 1" need to be edited as "We next elaborate on the mechanism of action of each cell type within the NVU in NVC, which displays selected representative studies of cellular mechanisms of NVC, mainly showing the cells involved in studying NVC, animal models, stimulation modalities, and imaging techniques in Table 1". The authors should explain what means "/" symbol in Table 1?

Response 1: Thank you very much for the valuable comments. We have reedited the sentence as suggested. (page 4,lines 150-153). In addition we explained the meaning of  "/"  at the bottom of Table 1. "/" indicates specific pathway studies or agent interventions not covered by the study. (page 7, line 163).

Comment 2:"Ach" should be edited as "ACh".

Response 2: Thank you very much for the valuable comments, we have modified "Ach" to "ACh" based on the comments.

Comment 3: "aECs" abbreviation should be used in the first mention.

Response 3: Thank you very much for the valuable comments. We have revised them accordingly and refined the full name in the first reference to aECs. (page 12, lines 395-396).

 Comment 4:"For example, sleep and circadian mechanisms appear to regulate barrier permeability, at least in invertebrate models."-"barrier" should be explained.

Response 4: Thank you very much for the valuable comments. The barrier here refers to the blood-brain barrier, which we modified in the manuscript. (page 22, line 863).

Reviewer 3 Report

Comments and Suggestions for Authors

While the reduction in overlap is a positive step, 25% is still a significant amount and raises concerns about the manuscript's originality. It is essential to ensure that the manuscript represents a novel synthesis of existing knowledge rather than reiterating previously published work. The authors should provide a detailed breakdown of the overlapping sections and explain how they have rephrased or reinterpreted these sections to contribute new insights. Further revisions may still be needed to ensure originality and reduce potential self-plagiarism or redundancy.

  • Provide a detailed breakdown of overlapping sections flagged in the iThenticate report and explain how these sections have been revised to ensure originality.
  • Further reduce overlap if possible, particularly in sections that summarize existing literature.

The authors have revised the manuscript, and some revisions are valuable, but the description still leans heavily on summarizing existing knowledge rather than critically analyzing the impact of these techniques. For example, the authors could delve deeper into how these techniques have shaped specific breakthroughs in NVC research or discuss their comparative advantages and disadvantages in clinical versus research settings. The manuscript could benefit from a discussion of emerging technologies and their potential for addressing current limitations.

Strengthen the critical analysis of imaging techniques, therapeutic approaches, and cellular interactions. Focus on providing new insights rather than summarizing existing knowledge.

Improve the integration of cellular interactions in pathological states by providing a more cohesive narrative that ties together the roles of various components in disease progression.

Comments on the Quality of English Language

While the revisions may have improved language quality, a thorough proofread by a professional editor is recommended to ensure consistency and eliminate any remaining errors.

Author Response

Comment 1:While the reduction in overlap is a positive step, 25% is still a significant amount and raises concerns about the manuscript's originality. It is essential to ensure that the manuscript represents a novel synthesis of existing knowledge rather than reiterating previously published work. The authors should provide a detailed breakdown of the overlapping sections and explain how they have rephrased or reinterpreted these sections to contribute new insights. Further revisions may still be needed to ensure originality and reduce potential self-plagiarism or redundancy. Provide a detailed breakdown of overlapping sections flagged in the iThenticate report and explain how these sections have been revised to ensure originality. Further reduce overlap if possible, particularly in sections that summarize existing literature.

Response 1: Thanks for the kind comments. According to the iThenticate weight check report, most of the overlap is the full name of the proper noun. Of course, there are some parts that are more similar to the descriptions in the original study, and we have modified them accordingly in the manuscript. The main modifications are: partly a change in the presentation of the description of the objective findings, and partly a re-summarization of the results of the study. As a result of our efforts, we have done our best to minimize the checking rate of manuscripts (15%). The main changes in the manuscript are as follows: page 8, lines 198-203; page 10, lines 290-292; pages 10, lines 301-307; page 10, lines 316-318; page 11, lines 332-336; page 11, lines 368-371; page 13, lines 443-445; page 13, lines 454-456; page 13, lines 465-469; page 17, lines 590-592; pages 18-19, lines 653-656.

Comment 2:The authors have revised the manuscript, and some revisions are valuable, but the description still leans heavily on summarizing existing knowledge rather than critically analyzing the impact of these techniques. For example, the authors could delve deeper into how these techniques have shaped specific breakthroughs in NVC research or discuss their comparative advantages and disadvantages in clinical versus research settings. The manuscript could benefit from a discussion of emerging technologies and their potential for addressing current limitations. Strengthen the critical analysis of imaging techniques, therapeutic approaches, and cellular interactions. Focus on providing new insights rather than summarizing existing knowledge. Improve the integration of cellular interactions in pathological states by providing a more cohesive narrative that ties together the roles of various components in disease progression.

Response 2: Thanks for the kind comments. In response to your suggestions, we have added a critical analysis of imaging technologies to the manuscript, describing advantages and disadvantages, as well as shortcomings in clinical translation (page 15, lines 541-552). A description of emerging imaging technologies has also been added in part 6. (page 22, lines 845-849). A section on interventions for NVC has also been added for perspective. (page 22, lines 816-828).

Round 3

Reviewer 3 Report

Comments and Suggestions for Authors

Thanks for addressing concerns raised previously. While the manuscript has improved, I noticed some additional points:

1. Suggest adding "A Narrative Review" to the Title of the mansucript for transparency and clarity.

2. The role of astrocytes in NVC remains debated, particularly regarding whether astrocytic calcium signaling is essential for functional hyperemia. The exact contributions of pericytes versus VSMCs in regulating capillary and arteriolar blood flow are not fully resolved. Please expand. 

3. The manuscript highlights potential therapeutic strategies, such as targeting astrocytic calcium signaling, endothelial NMDARs, or pericyte contraction. However, these approaches remain largely experimental and have not been validated in clinical trials. For example, while optogenetics and nanotherapeutics are promising, their feasibility, safety, and scalability for human use are not critically evaluated.

4. Certain points, such as the role of astrocytes and endothelial cells in NVC, are repeated multiple times across sections, leading to redundancy. The discussion of imaging techniques could be streamlined to avoid overlap with earlier sections.

5. While tables summarize experimental findings, the manuscript does not include quantitative comparisons of NVC impairments across diseases or treatments. For instance, how does the degree of NVC dysfunction differ between stroke and Alzheimer’s disease?

6. The manuscript provides an in-depth review of NVC mechanisms, its cellular components, and clinical implications in cerebrovascular disorders, but it overlooks the critical role of brain collateral circulation in maintaining NVC during cerebrovascular pathologies, which should be included along with recent advancements as discussed in previous work by Sinha et al. (Eur J Neurosci, 2024), Binder NF et al. (Neuron, 2024), and Ravindran et al. (Eur J Neurosci, 2021).

Comments on the Quality of English Language

The manuscript still needs significant editing for language and comprehension. Several sections have redundant and overlapping content.

Author Response

Comments and Suggestions for Authors

Thanks for addressing concerns raised previously. While the manuscript has improved, I noticed some additional points:

Comment 1 Suggest adding "A Narrative Review" to the Title of the mansucript for transparency and clarity.

Response 1: Thanks for the kind comments. Following this suggestion, we added “A Narrative Review” after the title.

Comment 2 The role of astrocytes in NVC remains debated, particularly regarding whether astrocytic calcium signaling is essential for functional hyperemia. The exact contributions of pericytes versus VSMCs in regulating capillary and arteriolar blood flow are not fully resolved. Please expand.

Response 2: Thanks for the valuable comments. In the chapter of Astrocytes and NVC, we proposed that the role of astrocytes in NVC is still controversial, and maybe our description is not clear enough. We have redescribed this section. (page 9, lines 243-245, lines 255 and 266).

The relevant role of vascular endothelial cells in the regulation of capillary and arterial blood flow is described in the subsection “Endothelial Cell and NCV”. The relevant role of pericytes in the regulation of capillary and arterial blood flow is described in the subsection “SMCs, Pericytes and NCV”. In addition, we have removed some duplicates from the manuscript. (page 12, lines 402-415).

Comment 3 The manuscript highlights potential therapeutic strategies, such as targeting astrocytic calcium signaling, endothelial NMDARs, or pericyte contraction. However, these approaches remain largely experimental and have not been validated in clinical trials. For example, while optogenetics and nanotherapeutics are promising, their feasibility, safety, and scalability for human use are not critically evaluated.

Response 3: Thanks for the valuable comments. We couldn't agree more with this. It is therefore emphasized at the end of the “Potential Intervention Targets and Strategies” according to the comment. (page 22, lines 824-826).

Comment 4 Certain points, such as the role of astrocytes and endothelial cells in NVC, are repeated multiple times across sections, leading to redundancy. The discussion of imaging techniques could be streamlined to avoid overlap with earlier sections.

Response 4: Thanks for the valuable comments. For the section on the duplication of the roles of astrocytes and endothelial cells in NVC, we have removed the relevant content from the Cellular Contributions to Neurovascular Coupling section. (page 3-4, lines 107-151). We have also removed some of the discussion on imaging technology to avoid repetition. (page 22, lines 842-846).

Comment 5 While tables summarize experimental findings, the manuscript does not include quantitative comparisons of NVC impairments across diseases or treatments. For instance, how does the degree of NVC dysfunction differ between stroke and Alzheimer’s disease?

Response 5: Thanks for the kind comments. Most of the experimental results in Table 1 are not based on disease models to assess NVC and are mostly studied by transgenic or knockout animals on specific targets. To our knowledge, few studies but manuscripts have quantified NVC injury and focused mainly on phenomenological studies. How much NVC damage affects cerebral blood flow and neurological function needs to be confirmed by further studies in the future. This is also mentioned in future research directions.

Comment 6 The manuscript provides an in-depth review of NVC mechanisms, its cellular components, and clinical implications in cerebrovascular disorders. Still, it overlooks the critical role of brain collateral circulation in maintaining NVC during cerebrovascular pathologies, which should be included along with recent advancements as discussed in previous work by Sinha et al. (Eur J Neurosci, 2024), Binder NF et al. (Neuron, 2024), and Ravindran et al. (Eur J Neurosci, 2021).

Response 6: Thanks for the valuable comments. We agree with this view and have made additions to emphasize the role of cerebral collateral circulation in NVC. (pages 13-14, lines 481-510).

“The role of cerebral collateral circulation in NVC is a complex and finely regulated process that involves a variety of factors and mechanisms to ensure that in the event of stenosis or occlusion of the main supplying arteries, blood flow is provided to the ischemic region via other vascular pathways to maintain perfusion and function of the brain tissue. In NVC, the role of cerebral collateral circulation is not limited to providing alternative blood flow pathways. It is also involved in the fine-tuning of neural activity.

Binder NF et al. revealed the role of Leptomeningeal collaterals (LMCs) in stroke reperfusion. They investigated this by laser scatter imaging, ultrafast ultrasound, and two-photon microscopy techniques in a thrombin-based mouse stroke model and found that LMCs maintained autoregulation of CBF and allowed gradual reperfusion, resulting in smaller infarct size. In mice with poorly functioning LMCs, distal arterial segments collapsed, and deleterious high blood flow after reperfusion led to hemorrhage and death. These findings suggest that LMCs are a key component in regulating reperfusion and preventing useless recanalization after stroke. The study by Ravindran et al. looked at the effects of cerebral collateral circulation in patients with acute ischemia undergoing reperfusion therapy. They emphasized the importance of cerebral side-branch circulation in stroke patients undergoing reperfusion therapy, particularly in predicting treatment outcomes. The results of these studies suggest that the state of the cerebral side branch circulation has a significant impact on successful revascularization and treatment outcomes.

The “Evolucollateral Dynamics” hypothesis proposed by Sinha et al. presents a novel framework for incorporating evolutionary biology principles into therapeutic strategies, providing new perspectives for enhancing collateral circulation. These studies suggest that the development and function of LMCs may be regulated by genetic predisposition and environmental influences, which have important implications for the development of therapeutic strategies and optimization of treatment outcomes. This hypothesis highlights the role of genetic predisposition and environmental influences on collateral circulation, which may influence treatment strategies and optimize therapeutic outcomes. Future studies must incorporate human clinical data to create robust therapeutic regimens that maximize the therapeutic potential of LMCs and improve the prognosis of stroke patients.”
